

# Large discrepancies between event- and response-based compound flood hazard estimates

Sara Santamaria-Aguilar[1,2], Pravin Maduwantha[1,2], Alejandra R. Enriquez[3], Thomas Wahl[1,2]

[1]Department of Civil, Environmental and Construction Engineering, University of Central Florida, Orlando, FL 32816, USA
[2]National Center for Integrated Coastal Research, University of Central Florida, Orlando, FL 32816, US
[3]School of Geosciences, College of Arts & Sciences, University of South Florida, St Petersburg, FL 33701, USA

*Correspondence to*: Sara Santamaria-Aguilar (Sara.SantamariaAguilar@ucf.edu)

**Abstract.** Most flood hazard assessments follow the event-based approach, assuming that the probability of flooding approximates the probability of flood drivers. However, this approach neglects information about the temporal and spatial variability of flood drivers and flood processes such as water propagation inland and its interaction with topography. The response-based approach accounts for these factors by using a large number of flood events that allow the calculation of flood probabilities. Here, we compare differences in flood hazards between the event- and response-based approaches for a case study in Gloucester City (NJ, U.S.). We find that compound events with return periods less than 20 years can produce the 100-year (i.e., 1% annual exceedance probability) flood depths in large areas of the city. This is caused by the temporal and spatial characteristics of these events, such as prolonged high coastal water levels and rainfall fields with higher rainfall rates over urbanized areas. These event characteristics are not included in extreme value models of the flood drivers and are commonly simplified by using a single design event. However, flood hazards largely depend on them, introducing large discrepancies in resulting flood hazards if neglected. The temporal and spatial variabilities of flood drivers need to be incorporated in flood hazard assessments to produce robust estimates.

## 1 Introduction

Coastal communities worldwide are facing increasing flood hazards from rising sea levels (Taherkhani et al., 2020; Wing et al., 2024) and extreme events such as tropical cyclones (Nederhoff et al., 2024). The rapid development in coastal zones compared to the hinterland is also contributing to increasing the exposure to flooding of people and assets (Cosby et al., 2024), making flooding the costliest hazard for coastal zones. In the U.S. alone, damages from tropical cyclones exceeded $1 trillion since 1980 and account for more than 50% of total disaster costs every year (NCEI, 2024). Therefore, developing adaptation and mitigation strategies to reduce flood impacts and increase the resilience of coastal communities is essential.

The most common framework for estimating coastal flood risks is the one defined by the Intergovernmental Panel on Climate Change (IPCC) in the Special Report on Managing the Risks of Extreme Events and Disasters to Advance Climate Change Adaptation (SREX), in which risks are defined as a function of the hazard, exposure, and vulnerability (IPCC, 2012). In the context of coastal flooding, quantifying the likelihood of coastal flood hazards is thus the first step to estimating flood risks





and impacts. However, there is no standard approach to quantifying flood hazards, resulting in a variety of methods being used, and discrepancies between them not being well understood. There are two main general approaches to estimating flood hazards, namely event-based and response-based. The event-based approach is the most commonly used. It consists of

estimating first the probability of the flood driver(s), selecting one event of desired probability (e.g., 1% annual exceedance probability (AEP)), and assuming that the flooding resulting from that event approximates the occurrence probability of the event. This approach can be applied for the full range of probabilities of events, from low to high, and produce what is known as probabilistic flood hazard estimates (e.g., Kupfer et al., 2024). In the U.S., the event-based approach has been used and is recommended for the Atlantic and Gulf coasts by the Federal Emergency Management Agency (FEMA) to produce the 1%

AEP (or 100-year) flood elevations, which are used as a regulatory floodplain for management and planning (FEMA, 2022). However, selecting a single event that approximates the 1% AEP floodplain might not be a simple task. On the one hand, statistical extreme value models used to derive the likelihood of events focus only on the magnitude of the flood drivers, and the temporal and spatial variability during events are neglected (e.g., Jane et al., 2020; Moftakhari et al., 2019). In the case of coastal water levels, the lack of information about the temporal evolution of the event has been commonly simplified using

different approaches such as selecting one historical event as a "design event" and rescaling its time series to the desired magnitude, e.g., matching the 1% AEP event (Dawson et al., 2005; Peña et al., 2023; Wadey et al., 2015); assuming a triangular or sine shape (e.g., Vousdoukas et al., 2016, Moftakhari et al., 2019); or defining a mean hydrograph shape from hindcast data (Dullaart et al., 2023). However, neglecting the temporal variability of coastal water levels can introduce large uncertainties in estimated flooding (Kupfer et al., 2024a; Quinn et al., 2014a; Santamaria-Aguilar et al., 2017a). Furthermore, when flooding

results from multiple drivers (e.g., tropical cyclones producing both storm surge and heavy rainfall), various combinations of driver magnitudes may share the same probability yet lead to differing flood depths and extents (see e.g. Peña et al., 2023). On the other hand, flooding (i.e., the response) also depends on other factors beyond the flood driver characteristics such as topography and associated water dynamics.

In contrast, the response-based approach can account for all these factors to produce more robust flood hazard estimates. This

approach involves simulating flooding from many events, enabling the calculation of empirical flood depth distributions at different points in the floodplain. However, the response-based approach also has limitations. First, a large set of events is needed, which is unavailable in observed records that rarely span more than a few decades up to century (Ponte et al., 2019). Therefore, synthetic event datasets generated through dynamical modelling and/or complex statistical frameworks are necessary (Gori et al., 2020; Kim et al., 2023; Maduwantha et al., 2025a). Second, this approach is computationally more

demanding and hence it has been rarely used in the past, but it is becoming more feasible due to advances in computing power (Gori et al., 2020) and new computationally efficient flood models (Bates et al., 2005; Leijnse et al., 2021a). Although the response-based approach provides more robust estimates of flood depths at the household level (or at single points), the corresponding flood extent does not represent the floodplain of a single event, which might be needed for some applications such as emergency management, government budgeting for natural disasters, and insurance market. For floodplain

management and planning, FEMA only recommends the use of the response-based approach to produce the 1% AEP floodplain



of all Great Lakes coastal Flood Risk Projects and the Pacific Coast, where the wave runup is considered the main hazard and thus different combinations of sea states (i.e., wave climate) and water levels (i.e., combination of astronomical tide and storm surge) can lead to similar total water levels (FEMA, 2022). However, coastal flooding often occurs from a combination of different drivers such as storm surges, wave runup, tides, heavy precipitation, and river discharge; so-called compound events.

In fact, the risk of compound flooding from storm surges and rainfall is larger in the Atlantic and Gulf coasts of the U.S. (Wahl et al., 2015), for which FEMA recommends using the event-based approach. Although FEMA provides some guidelines to map the 1% AEP floodplain, Mapping Partners can deviate from the guidelines if they consider it appropriate (FEMA, 2022). Thus, choosing between an event-based or response-based approach to estimate flood hazard is a decision that can be made. However, this choice is challenging to make in advance since it is unclear how closely the 1% AEP event (in terms of the flood

drivers) approximates the 1% AEP flood (in terms of the response). To our knowledge, the differences in flood hazard estimates between these two approaches have not yet been evaluated, specifically when flooding arises from compound events.

Here, we explore how closely the 1% AEP event approximates the 1% AEP flood from compound events of precipitation and estuarine water levels in a case study for Gloucester City, New Jersey. We first assess the variability in flooding from different synthetic 1% AEP events of equal probability but different magnitudes, and temporal and spatial evolutions, to quantify the

uncertainties related to using a single design event for estimating flood hazards. Then, we compare flood extents and depths from the 1% AEP events with the response 1% AEP flood. Finally, we investigate which individual compound events can cause the response 1% AEP flood depth in different parts of the study area.

## 2 Study site

Our study site is Gloucester City, New Jersey, a small municipality located in the Delaware estuary (Fig. 1) frequently affected

by pluvial and coastal flooding (CDM Smith, 2023). We selected this study site based on the exploratory scoping analysis of Helgeson et al. (2024) for place-based convergence research. Gloucester City is bordered by water on multiple sides, with the Delaware River to the west and Newton Creek and Little Timber Creek to the north and south, respectively. The catchments of these two creeks are relatively small, extending slightly beyond the city's administrative boundaries and draining into the Delaware River to the north and south of Gloucester City. Alongside the confluence of Newton Creek, Little Timber Creek,

and the Delaware River, the city's low-lying terrain, with elevation <10 meters above NAVD88, makes it especially susceptible to compound flooding from rainfall and elevated estuarine water levels, including storm surges, tides, and river discharge.

The FEMA Risk Map and Report, dated in 1979 and updated in 2016, defines a coastal and riverine 1% AEP floodplain (i.e., Special Flood Hazard Area, SFHA) that covers large areas of the city, including five essential facilities (FEMA, 2016b). Until 2016, there have been five federal disaster declarations for flooding in Gloucester City, but only 118 properties have policy

coverage under the National Flood Insurance Program (NFIP). Gloucester City has been facing problems with repetitive localized pluvial and coastal flooding for years (Fig. S3 of Supplementary Material), further exacerbated by an inadequate





stormwater drainage system (Smith, 2023). This is also highlighted in the FEMA Risk Map, in which an intersection of the city outside the SFHA is marked together with a photo of flooding from an event in 2009 (FEMA, 2016a. Fig. S1).

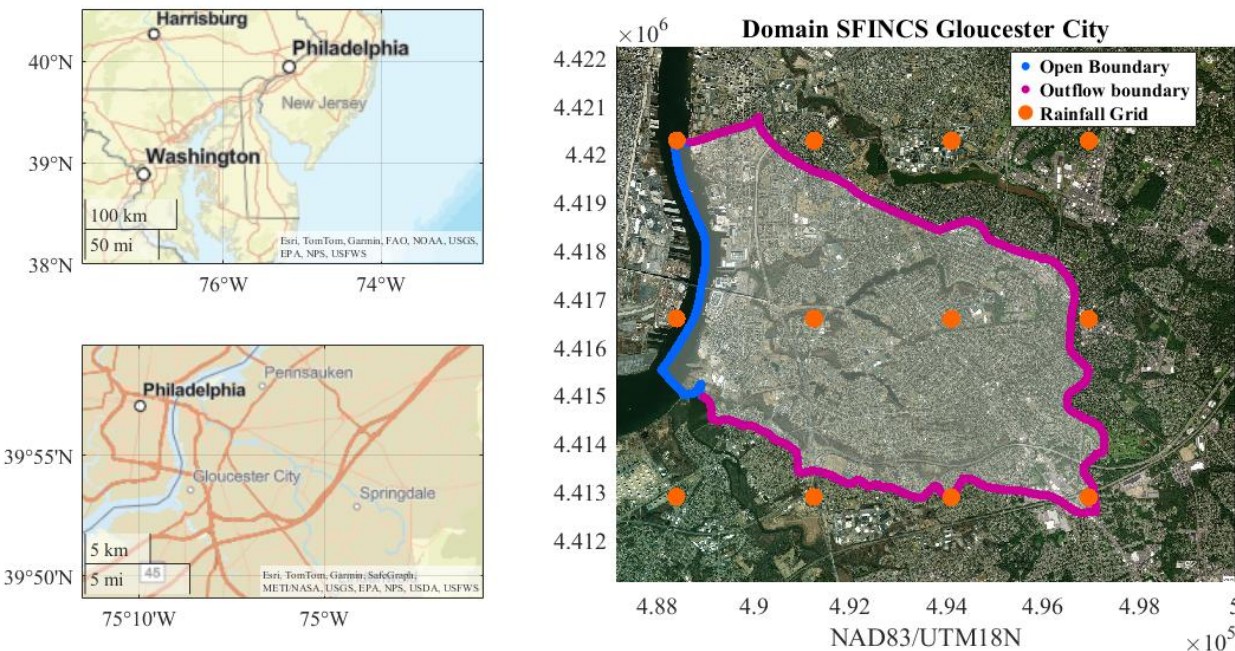

**Figure 1. Location of Gloucester City (NJ, U.S.) within the inner part of the Delaware estuary. The map on the right shows the flood model domain covering the catchments of Newton Creek and Little Timber Creek that surround the study site of Gloucester City. The blue line shows the location of the open boundary of the flood model along the Delaware River, the purple line is the inland outflow boundary, and the orange dots are the grid nodes of the rainfall forcing. [NAD83/UTM18N. ©Esri]**

## 3 Data and Methods

We investigate differences in flooding between the event- and response-based approaches by simulating flooding from a large number (5,000) of compound events that allow estimating the empirical distribution of flooding and comprise several 1% AEP events. We created a catalog of 5,000 synthetic compound events (more details on those events are provided in Section 3.1) following the framework of Maduwantha et al. (2025), which provides storm tide hydrographs and rainfall fields.

The joint probabilities of these events were calculated using the multivariate statistical framework of Maduwantha et al. (2024).

We use the reduced-complexity flood model SFINCS (Super-Fast INundation of CoastS) to simulate pluvial and coastal flooding from the synthetic events in the study area (Fig. 1). Details of the flood model configuration and input data are described in Section 3.2, and the model validation is presented in Section 3.3.

### 3.1 Synthetic compound events

We need a large sample of compound events to estimate the response-based flood hazard, in which the probability of certain
flood thresholds being exceeded is calculated for each model cell based on the empirical distribution. We use both the



multivariate statistical framework of Maduwantha et al. (2024) and the event generation approach of Maduwantha et al. (2025) to derive a catalog of synthetic compound events, including information of rainfall fields and coastal water levels along the Delaware River at Gloucester City. Maduwantha et al. (2024) developed a new multivariate statistical framework to estimate joint probabilities of rainfall and non-tidal residuals (NTR) accounting for the dependencies between these two flood drivers

but also stratifying the extreme events by the different storm types that generate them, namely tropical cyclones and non-tropical cyclones (Fig. 2), since these show different statistical characteristics. Non-tropical events dominate the low return levels, while tropical cyclones have a stronger effect on large return levels such as the ones associated with the 1% AEP event. Accounting for the different statistical characteristics of events caused by these different storm types, the joint probability analysis avoids mischaracterization of both low and high-return level events.

For the catchments of our study site, Maduwantha et al., (2024) used around 120 years of in-situ rainfall and coastal water level measurements to estimate the joint probabilities of the flood drivers, namely rainfall and NTR. They found that the largest dependency between NTR peak and rainfall exists for 18-hour rainfall accumulation. Since single-point rainfall might not be representative of the entire catchment, they also used 40 years of 4km gridded rainfall data from the Analysis of Period of Record for Calibration (AORC, Kitzmiller, 2018) of the corresponding catchments to obtain spatial rainfall information and

average catchment values. Further details about the multivariate statistical framework can be found in Maduwantha et al. (2024). Maduwantha et al. (2025) developed an approach to generate synthetic compound events based on the joint probability distribution from the previous analysis and by considering the temporal and spatial information of historical events. From the joint probability distribution, they derived a sample of 5,000 events ensuring that the proportion of observed tropical and non-tropical events is retained in the synthetic data (Fig. 2). Dynamic flood models such as SFINCS also require information about

the temporal evolution of events, namely time series of both coastal water levels and rainfall fields. For that, Maduwantha et al. (2025) used the time series of historical events to generate new time series for the synthetic event set. For each synthetic event, the time series of a historical event is selected randomly accounting for their proximity in the joint probability space, and thus accounting for differences in the temporal and spatial characteristics of these events depending on their magnitude. The historical event is then rescaled to the desired magnitude of the synthetic event. The rescaled NTR time series is then

combined with a mean sea-level value and a tidal curve while accounting for seasonality. The NTR hydrograph (i.e., time series) and the selected tidal curve are combined by selecting the lag from the observed events in order to account for the tide-surge interaction. Likewise, the synthetic rainfall field is combined with the synthetic water level hydrograph selecting a time lag between peaks based on the observed historical events. Further details about the methodology used to generate the synthetic compound events can be found in Maduwantha et al. (2025).






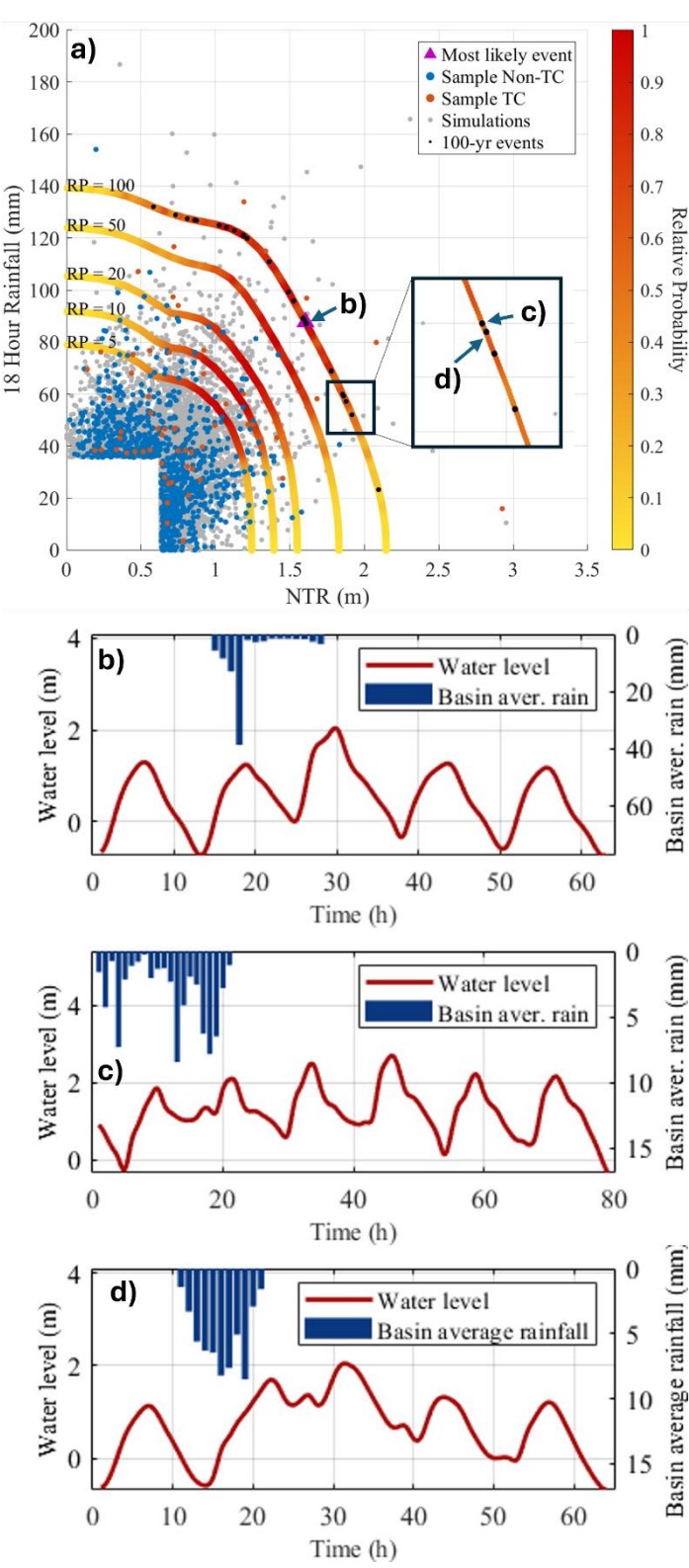





**Figure 2. a) Joint probabilities of non-tidal residual (NTR) and 18hr rainfall accumulation. Blue dots show the historical non-tropical cyclone events, red dots show the historical tropical cyclone events, and grey dots the 5,000 synthetic events generated for this study. All synthetic events (grey points) have assigned water level hydrographs and rainfall fields to be used as boundary conditions for**
**SFINCS. b) c) and d) show as an example the time series of three 1% AEP (100-year) events (black dots along the 100-year isoline); b) shows the time series of the "most likely" event, marked as a purple triangle in a); c) shows the 1% AEP (100-year) event that produces the largest flood from all 1% AEP (100-year) events (black dots in a); and d) shows the 1% AEP (100-year) event that produces the smallest flood from all 1% AEP (100-year) events. Water levels are referenced to NAVD88.**

Of the 5,000 synthetic compound events used, 25 lie along the 100-year (1% AEP) isoline (Fig. 2). Based on the density of

observed events along the isolines, we can define the relative probability of these events (Salvadori et al., 2011) to identify the

"most likely" event along the isoline.

The water levels at the Delaware River boundary of the model are also affected by the tidal variability, which is periodic and

thus its probability is not included in the multivariate extreme method of Maduwantha et al. (2024) for stochastic variables.

We estimate the likelihood of tidal levels based on the predicted tides of the 19-year period from 2003 to 2021 to include long-

term tidal variations such as the perigean and nodal cycles (4.4 and 18.6 years). Predicted tidal levels are generated based on

the annual harmonic analysis performed by Maduwantha et al. (2024) including nodal corrections estimated from astronomical

parameters (see Codiga (2011) for further information about the tidal harmonic analysis using UTide). We focus only on the

likelihood of high tide peaks since flooding is more likely at these levels, but it is important to notice that the synthetic events

are generated by combining the NTR peak and the high tide peak accounting for the historical distribution of time lags, and

thus accounting for tide-surge interactions (Maduwantha et al., 2025).

The tidal regime in our study region is mixed semidiurnal, with two high tides per day, but one is higher than the other. We

calculate the Mean Higher High Water (MHHW) level following the definition by the National Oceanic and Atmospheric

Administration (NOAA) to provide an average level of the largest tidal level that happens once a day. MHHW is estimated as

the average of the higher high water peaks of each day over a specified period, which in our case is the 19-year period from

2003 to 2021 (instead of the National Tidal Datum Epoch (1983-2001) used by NOAA), in order to provide an updated estimate

of MHHW and better representing present-day tidal conditions (Fig. 3). Although the largest variability of tidal levels is at

daily scale, tidal high waters also vary at fortnightly, seasonal, and interannual time scales. Therefore, we also estimate the

mean spring tidal levels as the average of the largest high-water levels every 14 days and the average "king tide" as the mean

of the annual largest tide over the 19-year period (Fig. 3).



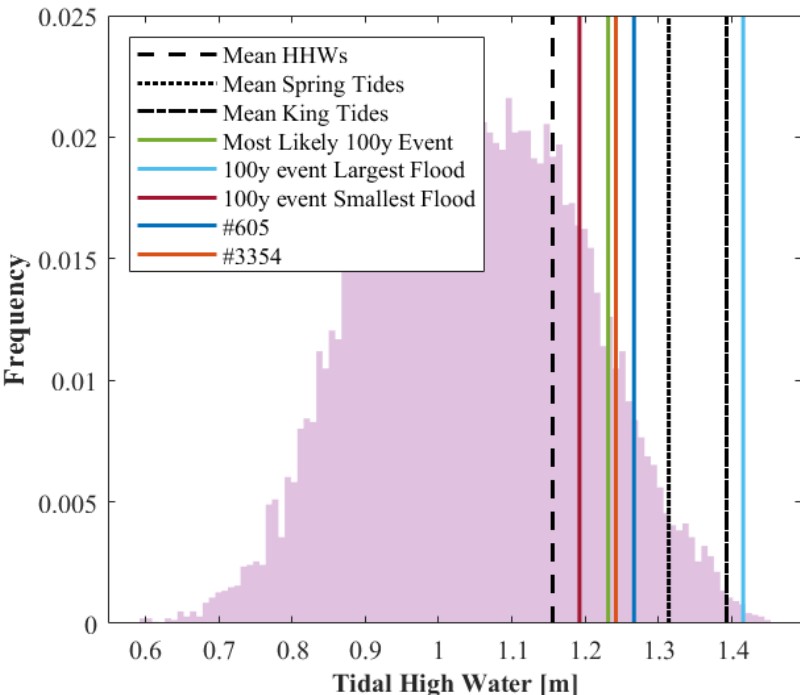

**Figure 3 Histogram of tidal high water levels over the last 19 year period from 2003 to 2021. Black lines show the tidal levels of MHHW, Mean Spring Tide, and Mean King Tide (defined as the largest annual tide). Coloured lines show as an example the largest high tide levels of the tidal curves selected for the synthetic events shown in Fig. 2b-d, and two additional synthetic events (#605 and #3354) discussed in the results section.**

Previous studies pointed to periods of increases in both high-tide flooding (Thompson et al., 2021) and extreme coastal flooding (Enriquez et al., 2022) caused by the nodal and perigean modulations of high-tide levels. Although these modulations are at longer time scales (4.4 and 18.6 years), the next peaks of both cycles will occur between 2025 and 2034 for diurnal and semidiurnal regimes. Since the tidal regime in our study site is mixed semidiurnal, the peaks of these two long-term tidal cycles are expected to occur within that period. To evaluate potential impacts of the long-term tidal modulations on the compound flood analyses, we estimate the 4.4- and 18.6-year tidal cycles following the approach of Enriquez et al. (2022) for the tide-gauge records of Philadelphia. We fit a least-squares regression to the annual king tidal levels (Eq. 1) of the last 60 years of record as suggested by Haigh et al. (2011).

$$H(t) = \beta + \beta_1(t) + \beta_2 \cos\left(\frac{2\pi}{4.4}t\right) + \beta_3 \sin\left(\frac{2\pi}{4.4}t\right) + \beta_4 \cos\left(\frac{2\pi}{18.6}t\right) + \beta_5 \sin\left(\frac{2\pi}{18.6}t\right) \tag{1}$$

Where H(t) are the king tides of each year $t$, $\beta_0$ is a constant term, $\beta_1$ is the linear term, $\beta_2$ and $\beta_3$ are the amplitudes of the perigean cycle and $\beta_4$ and $\beta_5$ are the amplitudes of the nodal cycle. Based on the fitted regression, we estimate the amplitudes



of both the perigean and nodal cycles, and the timing of the next peak of both cycles for our study region; this provides a better estimation of the present-day probability of large astronomical tides.


## 3.2 Flood Model

We use the dynamic flood model SFINCS (Super-Fast INundation of CoastS), which was designed specifically for simulating flooding from multiple flood drivers (Leijnse et al., 2021b), since we are interested in capturing interactions between rainfall and coastal water levels as well as the effects of spatio-temporal variability of compound events on the flood response. SFINCS

is a reduced-complexity flood model that balances computational efficiency with accuracy, making it a perfect candidate to simulate thousands of events at a reduced computational cost.

The municipality of Gloucester City is encircled by the catchments of Newtown Creek and Little Timber Creek, for both of which discharge data is unavailable. Therefore, we define the SFINCS model domain (Fig. 1) to cover the catchments of these two creeks by their 14-digit hydrologic units from the NJDEP Bureau of GIS (Table S1). This domain encompasses all runoff

that could potentially lead to pluvial flooding in the study area or fluvial flooding from the creeks. We define an open boundary along the Delaware River where the water level boundary conditions are given. The inland boundaries of the model domain are defined as "outflow" to allow any water flow to exit the domain. We use the subgrid approach of SFINCS with a dual resolution of 10 m and 1 m and the Digital Elevation Model (DEM) of Coastal National Elevation Database (CoNED) from the U.S. Geological Survey, which has a horizontal resolution of 1 m and a vertical accuracy of 10 cm (Danielson et al., 2016).

This DEM at 1 m is aggregated using the median to 10 m in ArcGIS pro-3.2.0. We use spatially varying surface roughness based on land cover data from the NJDEP Bureau of GIS (Table S1), converting land classifications into Manning's coefficients based on guidance from the U.S. Army Corps of Engineers (USACE, 2021). Water level boundary conditions are provided as time series at the location of the Philadelphia tide-gauge and the model interpolates them along the open boundary of the Delaware River. Rainfall forcing is applied as spatially varying fields, with the same resolution as the AORC data (Fig. 1);

SFINCS interpolates these onto the model grid resolution. The model is run with the advection term neglected, solving the local inertia equations. We use the GPU version of SFINCS and run the 5,000 simulations on an Intel(R) Core (TM) i7-13700KF CPU and NVIDIA GeForce RTX 4080 GPU. The outputs of the simulations at 10 m resolution are downscaled to 1 m resolution using MATLAB 2023a.

## 3.3 Flood Model Validation

Validation and calibration of flood models is a difficult task due to the common lack of observed flood data worldwide (Merz et al., 2024a; Molinari et al., 2019a). This is especially true for under-resourced regions; but the lack of observed flood data is also an issue in developed countries and more noticeable in the case of pluvial flood events, which are the most frequent in our study area of Gloucester City (Hino and Nance, 2021).





Different types of observations can be used to validate and calibrate flood extent and water depths, such as high-water marks and satellite images. In Table S2, we list the different sources we considered in this study to validate the flood model; it shows that little information is available about historical flooding in Gloucester City. Satellite images (or aerial images) are the only type of observed data that can measure the flood extent (Bates, 2004), and although these can be very useful for validating and calibrating large-scale flood models against large flood events (Tellman et al., 2021), for small-scale applications their

resolution is too coarse (Table S2, row 1). For validating and calibrating water depth outputs of flood models, high-water marks are commonly used. In the U.S., the U.S. Geological Survey, in partnership with FEMA, is the agency responsible for collecting high-water marks after major flood events (Table S2, row 2). However, no high-water marks from USGS are available for Gloucester City. Another source of flood-related information is the NOAA Storm Event dataset, which documents significant weather events from 1950 onwards. However, no flood events have been recorded for Gloucester City (Table S2,

row 3). Comparison against FEMA Special Flood Hazard Areas (SFHA) has also been used as a measure of flood model performance (Bates et al., 2021). However, we are including pluvial flooding, which is neglected when delineating the FEMA SFHA, and thus the comparison of our 1% AEP flood map to the FEMA SFHA will not provide a suitable measure of model performance. In addition, some studies have pointed to inaccuracies in the 1% AEP FEMA SFHA (Flores et al., 2023). Nevertheless, we found useful information for one particular flood in Gloucester City in the Flood Risk Map of FEMA (FEMA,

2016b; Table S2, row 4). This map is paired with a photo of a flooded intersection that is located outside of the SFHA and marked on the map with a star (Fig. S1). This flood happened in 2009, and based on the water level records and precipitation data, the flooding occurred due to a short but heavy rainfall event, with a maximum precipitation of 33.5 mm/h. We simulated this event using two approaches: one neglecting infiltration and another accounting for infiltration based on the Curve Number method included in SFINCS. We find that including infiltration underestimates flooding since that simulation results in no

flooding in the intersection where flooding was documented (see Pollack et al., 2024 for further details). Therefore, we decided to neglect infiltration in the configuration of SFINCS to perform all simulations in this study. Neglecting infiltration, the average water depth resulting from the simulation of that event in the intersection is 36 cm, a value similar to the water depth (approx. height of the curbside) shown in the FEMA Flood Risk Map (FEMA, 2016b; Fig. S1). We also searched for reported flooding in Gloucester City in local news, finding a couple of articles regarding a flood event that occurred in June of 2019

(Table S2, row 5). That event flooded a few areas of the city and led to power outages. In one of the articles, flooding of ~91 cm (3 feet) was reported in the basement of the Historical Society Museum. We simulated this event, finding that our flood model results in flooding of the area where the museum is located with average water depths of 68 cm (2.23 feet).

Since almost no information about historical flood hazards was found in the most commonly used sources, we also considered crowd-sourced platforms such as social media and the citizen science platform MyCoast: New Jersey. Crowd-sourced observed

flood data such as photos and geo-localized tweets have been shown to be useful for flood model validation as they often include information about the location of the flood and water depth levels can be extracted from photos (de Bruijn et al., 2019; Wang et al., 2018). We checked the Global Flood Monitor dataset (de Bruijn et al., 2019), which has an algorithm that automatically searches for flood keywords within tweets posted in any location of the world. In this database, we found nine





tweets reporting flooding within the region of interest. However, only one of them (which occurred in 2020) contained photos
and information about the exact location of the flooding reported (Table S2, row 6). In one of these photos, a traffic barrel is
shown in reference to the flooding, which allows us to estimate a flood depth of approx. 60 cm based on standard traffic barrels
of 5 sheeting of 15.24 cm each. We simulated that flood event and obtained average water depths of 72 cm around the streets
and intersections reported as flooded in the tweet (Fig. S2). The citizen science platform MyCoast (Table S2, row 7) allows
citizens to upload photos of flooding anywhere in New Jersey, but it does not have any photos of flooding in Gloucester City.
We also contacted local authorities to obtain any information available about historical floods in Gloucester City. In this
context, the Camden County Municipal Utilities Authority (CCMUA) has recently identified flood-prone locations in
Gloucester City as part of the first phase of a regional flood study to design flood mitigation measures for the region (Table
S2, row 8 and Fig. S3). We compared the 10-year return period flood map to these geo-located regions to evaluate whether
these known flood-prone locations experience flooding from events with a relatively short return period. We found that the
10-year return period flood map captures those flood-prone locations well (Fig. S3).

Although the limited observed flood data for Gloucester City did not allow for a quantitative flood model validation, we believe
that the qualitative validation performed based on the simulation of the events of 2009, 2019, and 2020, together with the
evaluation of the known flood-prone regions represented in the 10-year flood hazard map, provides the best possible flood
model validation for our study site.

## 4 Results

### 4.1 Differences in flood response from events with the same (joint) probability

We simulate flooding from 25 events with a 1% AEP that have different combinations of the magnitude of rainfall and NTR
peak, but also different temporal and spatial evolutions and are combined with different tidal curves (Fig. 2). We find that the
floodplain of each of these 1% AEP events is different, resulting in very large differences in both flood extent and depth
between some of the events. In Figure 4-a, we show the frequency of flooding at each cell ($1m^2$) from all 25 1% AEP events;
a frequency of 1 indicates that the particular area is flooded from all 25 events and a frequency near zero indicates that this
area is only flooded from one or few of the 25 events. Certain areas scattered throughout the municipality experience flooding
during all 1% AEP events (green areas in Fig. 4-a). However, a larger area is flooded only by a few of the 1% AEP events,
which shows that selecting only one 1% AEP event for estimating flood hazard can introduce large uncertainties in exposure
(and subsequently risk). Areas flooded only from a few 1% AEP events are mainly along the Delaware River and creeks, where
both flood drivers interact. In contrast, pluvial hot spots, i.e. regions that are not hydrologically connected to the Delaware
River or creeks and thus rainfall is the only flood driver, show almost no variability in flood extent between the different 1%
AEP events.




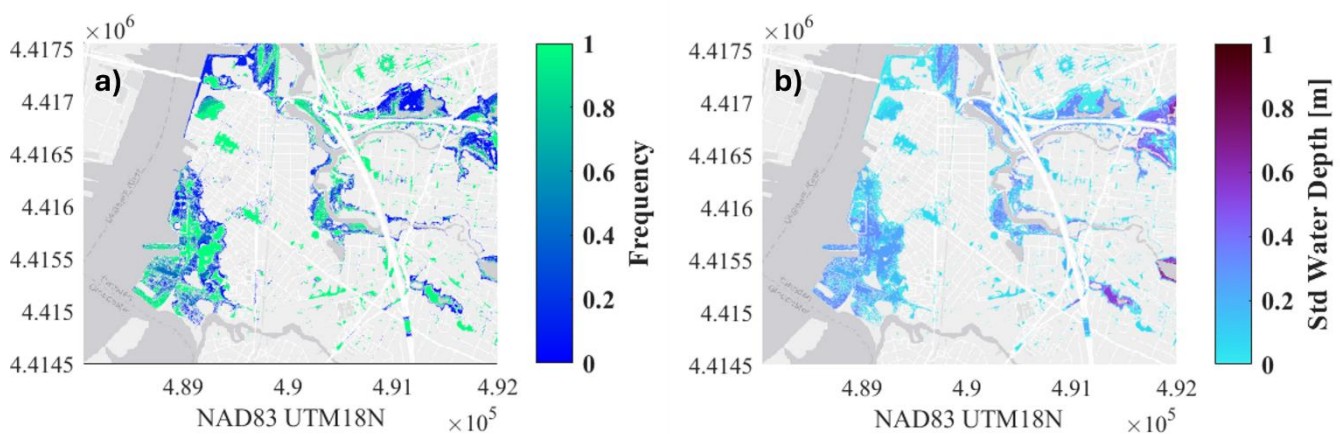

**Figure 4 a) Frequency of flooding at each model cell (1m$^2$) from the 25 events selected along the 100-year (1% AEP) isoline. b) The standard deviation of water depth (m) at each model cell (1m$^2$) between the 25 events selected along the 100-year (1% AEP) isoline. [NAD83/UTM18N. ©Esri]**

We also analyze the variability in water depths using the standard deviation between the flooding from all 25 1% AEP events in all model cells (Fig. 4-b). Larger standard deviations in water depth exist in regions where all 1% AEP events produce flooding, with maximum values of ~0.8 m. Larger variability of water depths also exists in coastal regions, where both flood drivers interact, while small variations occur in pluvial hotspots.

The largest and smallest flooding, in terms of flood extent and volumes, are produced by events that have almost the same 18h

accumulation rainfall and NTR peak, 59.18 mm and 1.86 m and 57.18 mm and 1.88 m respectively, and thus lie very close to each other on the 100-year (1% AEP) isoline (Fig, 2 c and d). However, the NTR hydrographs of these events are different, with one of them lasting for several hours with sustained large water levels (Fig. 2c) while the other is shorter and with lower water levels (Fig. 2d). In addition, the two events are combined with different tidal curves with high-tide levels that differ by more than 20 cm, an average MHHW and a larger than average king tide (see Fig. 3). The 1% AEP event that produces the

smallest flooding is combined with a tidal curve with high-tide levels similar to MHHW, but the 1% AEP event that generates the largest flood is combined with a tidal curve that reaches values larger than the average king tide (Fig. 3). These factors cause the water level hydrographs of these events to differ in their temporal evolution and to have water level peaks that differ by ~0.5 m, which combined leads to the large differences in flood response. Although the tidal curve combined with the 1% AEP event that produces the largest flood might appear "extreme", the analysis of the long-term modulations of the tide reveals

that this king tide level was reached several years earlier in the current nodal cycle (Fig. 5). By extending the fitted long-term modulation, we show that the tides are currently in the ascending phase of both nodal and perigean cycles, with a peak expected in 2026. As a result, the likelihood of the tidal level of this -particular synthetic event is higher over the coming years.



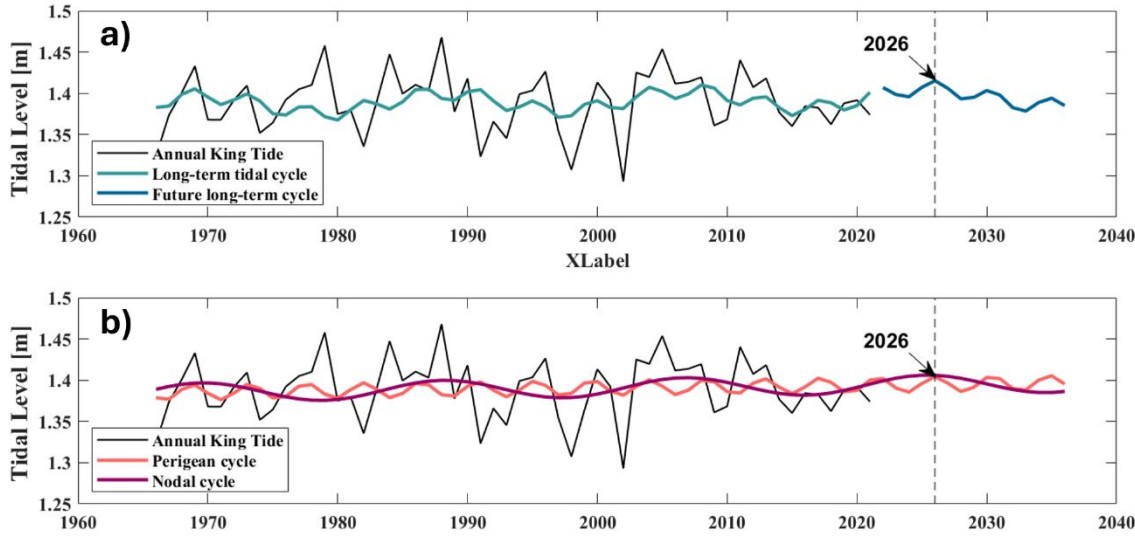

**Figure 5 Annual king tide levels (as the maximum tidal level) at the tide-gauge of Philadelphia used in this study. Estimated long-**
**term variability of king tides (a) from the combined nodal (18.6-y) and perigean (4.4-y) cycles including prediction of future**
**combined peak of these long-term cycles expected for 2026; (b) separated nodal (18.6-y) and perigean (4.4-y) cycles for the historical**
**and future period.**

These results show that the variability of the NTR hydrograph, together with the variability of the tidal curve, have very large
effects on the resulting flooding since events with almost equal NTR peaks can produce very different flooding. The
topography also plays an important role; when the water level at the Delaware River boundary exceeds the elevation of the
coastline, the large low-lying region behind it floods. Thus, small increases in water levels along the hydrograph can cause
large changes in flood extents when certain thresholds are exceeded.

We also compare the flooding arising from all 1% AEP events with the "most likely" event in order to assess the uncertainties
related to the use of a single design event when assessing flood hazards. This is commonly done when following the event-
based approach. In terms of flood extent and volumes (Fig. 6), most of the 1% AEP events (17 and 19 of the 25, respectively)
produce larger flooding than the "most likely" event. However, there are substantial spatial variations between events, as some
can produce larger flooding in some areas and smaller flooding in other areas (not shown). We calculate the total flood extent
and volume of all 5,000 events to estimate the empirical return periods of these two flood metrics (Fig. 7). Based on the
empirical distribution, the "most likely" event has a return period of 38 years in terms of extent and 33 years in terms of total
flood volume. 20 of the 1% AEP events have return periods <100 years (> 1% AEP) in terms of total flood extent, while only
12 events have return periods <100 years in terms of total flood volumes. This shows that using a single design event when
assessing flood hazards can lead to large uncertainties in both flood extent and depth, often resulting in an underestimation of
the extent in our case study.




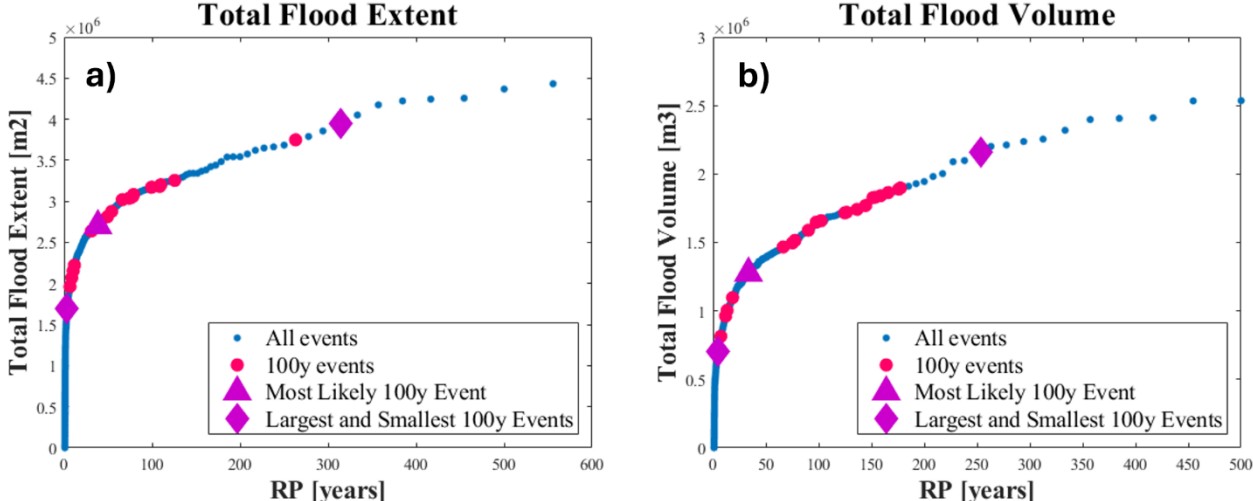


**Figure 6. (a) Empirical distribution of total flood extent from all events (blue points), 1% AEP events (pink dots) including the "most likely" (purple triangle) and largest and smallest flood (purple diamonds); (b) same as (a) but in terms of total flood volumes.**

### 4.2 Response-based flood hazard

We estimate the response-based flood hazard map by calculating the empirical distributions of water depths at each model cell
(of 1 $m^2$) and show the water depth corresponding to the 1% AEP (Fig. 6). This 1% AEP response flood hazard can thus be produced by different events in different regions. Comparing the response flood hazard to the flood hazard of the different 1% AEP events, the response flood hazard has generally larger flood extents and water depths, with a few exceptions. In the 1% AEP response flood hazard map, there is a larger area in the south of Gloucester City facing the Delaware River (Fig. 6) with water depths up to 1 m. However, this region is only flooded by a few 1% AEP events (Fig. 4). In contrast, the flood hazard
in the northern Delaware region from the response-based flood hazard map is similar to the region flooded by all 1% AEP events. In this region, the 1% AEP event that causes the largest overall flooding (Fig. 2c) also produces more extensive flooding. This might be caused by the relatively longer hydrograph of that event combined with a larger-than-average king tide. Comparing the differences between the response-based and event-based flood hazard in the pluvial hotspots (areas not hydrologically connected to the Delaware River), only one 1% AEP event causes larger flooding than the response-based
approach. In the northeast region of the domain, ten of the 1% AEP events produce larger flooding than the response-based 1% AEP floodplain. This can point to effects of the spatial variability of rainfall fields between events, which are masked in the joint probabilities because these are based on the 18h accumulated average rainfall in the catchment.



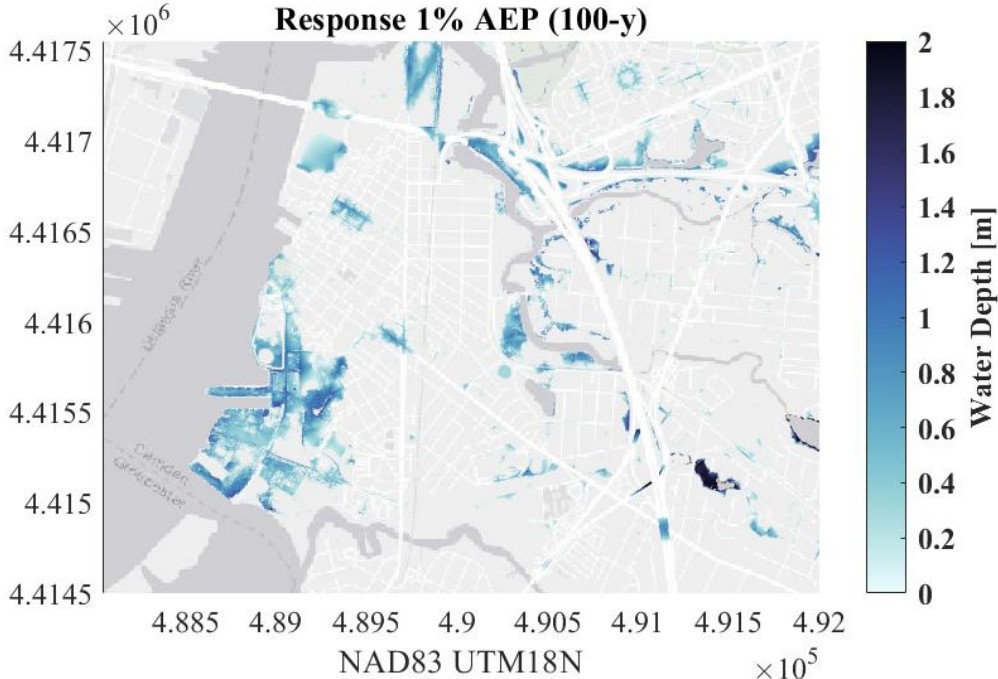

**Figure 7. 1% AEP response flood hazard map calculated by the empirical distribution of water depths from 5,000 simulations; we show the 1% AEP water depth at each model cell (1m²) [NAD83/UTM18N. ©Esri]**

We trace the events that produce the 1% AEP response-based flood hazard and group them by their corresponding return periods based on the joint probabilities of flood drivers (Fig. 8). This helps identifying the types of events causing the 1% AEP water depth in different areas of our study site. Across most of the study domain, and especially in the urbanized region in the centre of the city, events with joint AEPs much higher than 1% can lead to the 1% AEP water depth as identified from the response-based approach. Most of the 1% AEP water depths along the south coastal region of the city are produced by a single compound event with an AEP of ~50% (2-year return period; yellow) and another event with an AEP of ~7% (14-year return period; light orange) based on the joint probability distribution of NTR and 18h accumulated rainfall. Although the NTR peak of these events is around 1 m, and thus much smaller than other events, these two events have long hydrographs with sustained water levels for several hours and combined with tidal levels of around 1 m can produce the 1% AEP water depths in that area (Fig. S4 of Supporting Material). The tidal levels of these two events (#605 and #3354 in Fig. 3) exceed the MHHW but remain below the mean spring tides, making them likely to occur on a fortnightly basis. The 1% AEP water depths in regions that are only affected by pluvial flood events are generally also caused by events with AEPs >2%. Notably, there are two pluvial hotspots in the city region produced by events with AEPs >10% (less than 10-year return period). These are produced by two different events, both with AEPs of ~12% based on the joint probability distribution. More detailed assessment of the rainfall





fields of these events reveals that they have larger rainfall over that area of the model domain, which gets masked when
       averaging the rainfall over the entire catchment (Fig. S5-S6).

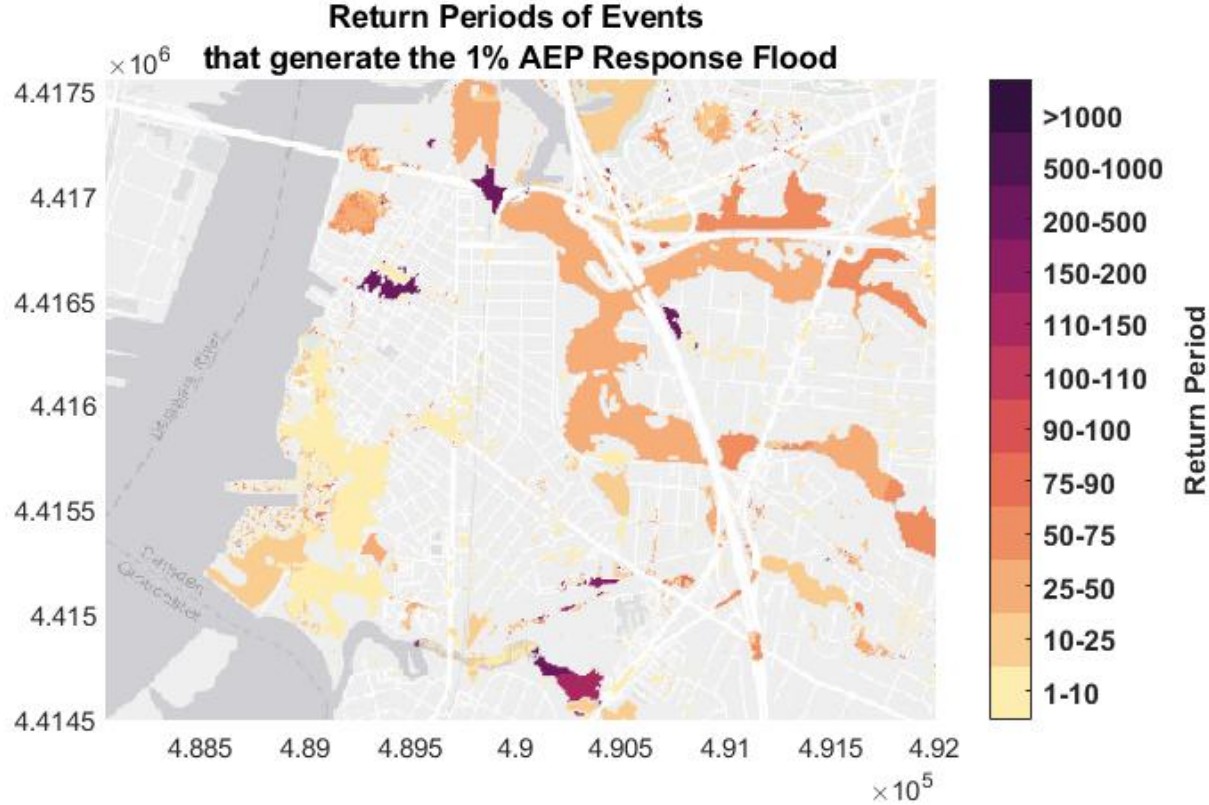

**Figure 8. Return periods (based on the bivariate extreme model) of the events that generate the 100-year water depth from the**
**response approach. [NAD83/UTM18N. ©Esri]**

**5 Discussion**

Lots of research has been dedicated to improving extreme statistics of compound events and to quantifying the uncertainties
of extreme value analysis of flood drivers (e.g., Lucey and Gallien, 2024) assuming that the probability of the event
approximates the probability of the resulting flooding. However, little research has been focused on analyzing the latter,
specifically for compound flooding, in which more than one driver is involved, and thus different combinations of flood driver
magnitudes have the same joint probabilities. Here, we have assessed how well the 1% AEP event approximates the 1% AEP
flood for a case study in Gloucester City (NJ, U.S.) by comparing the 1% AEP flood hazard based on the event- and response-
based approaches. We find that the 1% AEP water depth can be produced by different events in different parts of the city and
that the AEPs of these events are often much larger than 1%. Specifically, the region of Gloucester City with the largest 1%
AEP flood hazard is the coastal zone and it is caused by events with AEPs >5% (less than 20-year return period) based on the





joint probability distribution of 18h rainfall accumulation and NTR peaks. However, these events (Fig. S4) exhibit sustained high NTR levels, which, when combined with tidal levels larger than MHHW, can result in greater water depths than most of the analyzed 1% AEP events. Similarly, the regions impacted mainly by pluvial flooding also tend to experience the 1% AEP water depths from events with AEPs >10% (less than 10-year return period). In this case, the events producing the 1% AEP

water depths in the city show spatial variations in their rainfall fields, with larger precipitation rates over the urbanized city region. However, the spatial variability of the rainfall fields is smoothed when calculating the average rainfall in the catchment to perform the extreme value analysis, and together with small NTR peaks, these events get assigned high AEP values (or short return periods).

These results show that the response-based approach leads to better representation of flood hazard at the household level. It

accounts for the temporal variability of NTR hydrographs, combination with tides and mean sea level, and the spatial variability of rainfall fields. All of those are not explicitly accounted for in extreme value models to derive joint AEPs or joint return periods for the event-based approach. Nevertheless, some applications, such as emergency management, might need event-based flood hazard maps.

Event-based flood assessments commonly use a single design event, thus neglecting temporal and spatial variations of the

flood drivers along the duration of the event. We have shown that using a single design 1% AEP event can introduce large uncertainties in both flood extents and water depths that arise from the different combinations of the drivers' magnitude but are mostly due to the temporal and spatial variability of events. Events of almost equal magnitude but different spatial rainfall fields and temporal variability of the water level hydrographs can produce very different flood extents and water depths. The disparities in resulting flooding are more pronounced in the coastal areas of our study domain, where both flood drivers interact

and are further influenced by changes in tidal variability. Considering the variability of the tide, rather than relying on a single MHHW level is also crucial, as tidal fluctuations over longer time scales (such as spring and king tides) can influence coastal flooding. This is especially relevant now, as tides are in the ascending phase of their long-term cycles, which are projected to reach their peaks within this decade, with the first peak expected as soon as 2026 at our study site. This finding highlights the necessity of taking into account the variability in the tidal levels.

Ignoring the spatio-temporal variability of extreme events by relying on a single design event can lead to significant uncertainties in flood exposure, which in turn can result in substantial uncertainties in flood risks. One way to address this limitation when using the event-based approach is to employ ensembles of events that account for the spatial and temporal variability of the flood drivers. From that one can produce an ensemble of flood hazard maps for a desired return period, similar to probabilistic flood maps but for a given AEP (or return period). Our study has several limitations that highlight areas for

further research. We focused on a small study site with a particular topography that is affected by two flood drivers with associated variabilities. Thus, the results cannot be extrapolated to other coastal regions. However, we expect that our general conclusions are transferable to other regions. For example, the importance of temporal and spatial variations of the flood drivers has been pointed out by other studies in Germany (Kupfer et al., 2024b; Santamaria-Aguilar et al., 2017b) and in the UK (Quinn et al., 2014b), showing that changes in water level hydrographs can produce large changes in flood hazards.





Another limitation of our study is that we use a synthetic event set developed using a data-driven statistical framework, which is limited to observed events. Consequently, it may not fully capture the full range of the potential spatio-temporal variability of flood drivers. Tropical cyclones might also be underrepresented in the historical sample since their frequency of occurrence is very low. This limitation can be overcome by using synthetic tropical cyclones that are dynamically downscaled to the study site; this is computationally expensive but has been done for other regions (e.g., Gori et al., 2020). This allows for increasing

the number of extreme events in the sample but also for changes in the spatio-temporal variability of these events compared to the historical record.

Our flood model approach also has some limitations. First, we are neglecting the stormwater system and thus we might overestimate flooding and neglect the fact that some areas might experience flooding due to backwater effects in the system. Stormwater systems are typically designed for events with low to moderate return periods (often the 10-year event, or 10%

AEP event) while we have focused on the 1% AEP (100-year) flood hazard. Such events would likely exceed the capacity of the stormwater system. In addition, we neglected infiltration based on the validation of the model for one single event for which we have information on reported flooding at a single location. Although most of our study domain is urban and thus covered by impervious surfaces, we might underestimate infiltration in areas of the with larger amounts of vegetation. Both the validation of the flood model and calibration of parameters and processes such as infiltration can be improved if more

observed flood data from past events is available. The lack of this data is a common problem worldwide (Merz et al., 2024b; Molinari et al., 2019b) and it can be overcome by systematically collecting flood data after flood events or making available datasets such as claims from the National Flood Insurance Program (Sebastian et al., 2021).

## 6 Conclusions

Coastal communities are experiencing growing flood hazards due to rising sea levels, more frequent extreme events, and an

increase in population and assets in flood prone areas. Consequently, more robust flood hazard estimates are required to develop effective adaptation strategies to mitigate flood impacts. Although significant attention has been focused on reducing uncertainties in the estimation of probabilities of flood drivers, little is known about how well the probability of compound events approximates the probability of flooding. Here, we addressed this issue by comparing flood hazard derived from the event- and response-based approaches for a case study in Gloucester City (NJ, U.S.), which is frequently affected by pluvial

and coastal flooding.

Our findings reveal that the 1% AEP flood hazard derived from the response-based approach can be caused by different events in various parts of the city, with AEPs much larger than 1% (return periods <100-year). In the coastal area, events with AEPs >5% (less than 20-year return period) can produce a 1% AEP water depth if the NTR hydrograph leads to prolonged high water levels when combined with tidal levels between the MHHW and average spring tides. In this context, our findings are

in line with previous studies that highlighted that the long-term variability of tides can modulate both minor and extreme flooding (Enriquez et al., 2022b; Thompson et al., 2021b). We find that considering tidal variability is crucial, rather than





relying on the assumption of a constant MHHW, as flooding from both low and high return period events can differ substantially depending on the tidal level considered. Tides are currently in the ascending phase of the nodal and perigean cycles, which are expected to peak in 2026 in our study region, making it more likely that storm surges coincide with high tide
levels, thus increasing the probability of flooding. Similarly, not accounting for spatial variability of rainfall fields, which is masked when using catchment average values for extreme value analysis, can underestimate pluvial flood hazards. This study highlights the importance of considering temporal and spatial variability in flood hazard estimates. The traditional method of using a single design event in event-based assessments can lead to considerable uncertainties in flood extent and water depth, especially due to varying combinations of flood drivers. The response-based approach, which accounts for factors like tidal
variations and spatial variability in rainfall, provides a more robust representation of flood hazards. However, event-based maps remain essential for some applications such as emergency management. Using ensembles of events that account for these variations would enhance flood hazard estimates derived from the event-based approach.

While our results are not directly applicable to other regions, we expect similar conclusions elsewhere regarding the impacts on compound flood hazards from neglecting the temporal and spatial variability of flood drivers. Future work should focus on
producing more robust flood hazard estimates by using many compound events including their temporal and spatial evolution rather than focusing on single design events for given AEPs or return periods. Similarly, future projections of flood hazards should also account for potential changes in the temporal and spatial evolution of events rather than focusing only on changes in their magnitude.

**Code availability**

The SFINCS model is available at https://sfincs.readthedocs.io/en/latest/example.html#executable. The codes used for these analyses are available on GitHub (https://github.com/CoRE-Lab-UCF/MACH-Compound-Flooding/tree/main/Santamaria-Aguilar_et_al_2025_Event_Response) (The DOI and the final version of the codes will be available after addressing the reviewers' comments and suggestions.)

**Data availability**

The hydrologic units are available https://gisdata-njdep.opendata.arcgis.com/datasets/02599a9424254a4ea33e689941559e3c_17/explore. The DEM is available at https://www.usgs.gov/special-topics/coastal-national-elevation-database-applications-project/data, and land cover data is available at https://gisdata-njdep.opendata.arcgis.com/documents/njdep::land-use-land-cover-of-new-jersey-2015-download/about.
The SFINCS model files generated and used in this study are available at https://zenodo.org/records/14251309, and the 5,000 flood simulations are available at https://zenodo.org/records/15047845.



**Authors contribution**

The study was conceived by TW and SSA. SSA performed the simulations, undertook the analyses, and wrote the first draft of the paper under the guidance of TW, PW, and ARE. All authors co-wrote the final paper.

**Competing interests**

The authors declare that they have no competing interests.

**Acknowledgments**

S.S.A, P. M. and T.W. were supported by the National Science Foundation as part of the Megalopolitan Coastal Transformation Hub (MACH) under NSF award ICER-2103754. This is MACH contribution number 76. We would like to acknowledge
Maarten van Ormondt for providing technical assistance with the SFINCS model. We would also like to thank Stefan Talke for his support in the analysis of tidal levels.

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
