# Peer review of "Large discrepancies between event- and response-based compound flood hazard estimates"

_EGUsphere, 2025_

## Referee Comment (RC1)

**Summary**

This study compares the flood hazard estimates generated using an event-based approach to those generated using a full suite of synthetic events, referred to herein as a response-based approach. Using Glouchester, NJ as a case study, the researchers estimate the compound hazard resulting from the joint probability distribution of 18-hr total precipitation and non-tidal residuals. From this distribution, they generate 5,000 synthetic storm events, each of which has a statistically derived spatiotemporally varying rainfall field and storm tide hydrograph (non-tidal residual + tide). From these, the authors sample 25 synthetic storm events with an Annual Expected Probability (AEP) of 1% (i.e., 'event-based') sampling from the isoline (representing maximum likelihood) and compare the estimated 1% AEP inundation map generated using the entire suite of synthetic storm events (i.e., 'response-based'). They find that the response-based approach generally generates higher hazard estimates, suggesting that selecting the 1% events based on boundary conditions alone does not generate the 1% hazard.

Overall, I found this study very interesting to read and many of my comments and questions come from a place of genuine interest in the modelers' choices. My major review comments are as follows:

First, while the experimental design itself is not particularly novel, the combination of statistically-derived synthetic storm events that can represent compound drivers (both non-tidal residual and precipitation) as inputs to a process-based model takes many of the concepts that have been discussed within the compound hazard modeling community a step further than previous studies, and thus I find that it is worthy of publication.

Second, while this work is new in the context of compound flooding – a topic of considerable interest – the authors rely heavily on the compound flood literature. In doing so, they overlook a larger body of research regarding stochastic flood hazard simulation for rainfall-induced floods. For example, there have been several other studies that draw similar conclusions about the use of design storms vs stochastically generated (see, e.g., Perez et al. 2024, several papers by Daniel B. Wright). The paper could be further improved by placing itself in this body of work, perhaps as part of both the introduction and discussion.

Third, I commend the authors on their work to validate the hydrodynamic model in a data-scarce environment, however, I would like to see more discussion of the potential limitations of the synthetically generated boundary conditions. The proposed approach also introduces considerable uncertainties into the hazard estimates that are difficult to disentangle, including questions about how the authors are accounting for different flood types (e.g., those driven by high river events vs coastal storms) and whether the statistically-derived framework can adequately account for these.

The comments below primarily ask for additional clarification or suggest that the authors contextualize some of their findings.

**Comments**

Line 34 the authors differentiate between event-based and response-based approaches. Is 'response-based' a commonly used term in flood hazard literature? Is there a citation that can be used to support this terminology? I have often heard them differentiated as 'design-storm' vs 'probabilistic' or 'stochastic' methods. Is there a citation that can be included to help support this choice of terminology?

In this same paragraph there is some discussion of the FEMA regulatory maps and process used for flood hazard delineation in the U.S. It is perhaps necessary here to differentiate from the approaches used for inland and coastal hazard. While they both rely on event-based approaches, the inland hazards are derived from design storms and assume a one-to-one relationship between the AEP of the precipitation event

(spatially uniform, but varied in time according to a characteristic distribution) and the resulting flood flow which is then translated into hazard. In line 43, you state that temporal variability for inland flooding is neglected, but this is not true in the case of rainfall. It is true, however, in the case of flood mapping, where the peak flow is used to estimate the extent of the floodplain. There is some nuance here that gets lost in the way it is written now, and it might be worth revisiting this section.

With respect to coastal flood hazard estimation, I'm not sure I agree with what is written in lines 43-49. Perhaps you are referring to how boundary conditions are applied to the downstream end of an inland model(?), but with respect to the FEMA SFHA (V-zone), it is my understanding that the most recent version is derived from ADCIRC simulations (where events are created using JP-OMS) in which case the spatial and temporal variability of the drivers and also the resulting water levels are considered when estimating the resultant hazard.

Line 57 remove 'up to a century'

Line 74-76 I think you are correct that this has not been done for compound flooding; however, I would point out two relevant studies that undertake a similar analysis applied to rainfall flooding (one that uses SFINCS) and reach the same conclusion. I think you may want to point to these both in your introduction (and draw differences between your work from theirs) and again in the conclusion.

Perez, G., Coon, E. T., Rathore, S. S., & Le, P. V. (2024). Advancing process-based flood frequency analysis for assessing flood hazard and population flood exposure. *Journal of Hydrology*, *639*, 131620.

Baer, J. A. (2025). *Design Storms Underestimate Flood Hazard and Risk Derived From Stochastic Storm Transposition* (Master's thesis, The University of North Carolina at Chapel Hill).

Line 88-90 What is the total size of the model domain in km$^2$?

Line 93 Provide date range for the start of the federal disaster declarations, e.g., 'Between X and 2016, there were five federal disaster declarations…'

Line 94 Please provide more information about the building stock for context. You state that there are only 118 properties with coverage. Is this reflective of active policies based on OpenFEMA or where does this number come from? Do you also have the ability to provide a denominator for the total number of properties in Glouchester City and the number located within and outside of the FEMA SFHA? These would be useful numbers for context, given that insurance is only required at properties with a federally-backed mortgage within the SFHA.

Line 116-120 The downstream boundary condition at the Delaware River is assumed to be represented as a coastal water level? How do you consider the possibility of high river flow events (antecedent conditions that are not coastally driven) in your compound events framework? Have the previous works that are cited already established the relative frequency of high river flood events vs those that are driven by high coastal water levels and local rainfall? Do these events come from a different distribution than those that are considered 'compound coastal' events?

Section 3.3 You invested considerable effort in validating the model, which I think was excellent given how limited the data for this cite is, however, the process used to generate the synthetic events for the model also has many layers of uncertainty in it (e.g., statistically derived joint probabilities of precipitation and non-tidal residual, statistically derived spatio-temporal rainfall structures, scaled event magnitudes, lag). To what extent could your model framework allow you to explore some of these uncertainties? Is there any mechanism that you could use to test the compound events framework and

whether you believe your hazards estimates are true (assume the process-based model results are valid)? Given that you are comparing the model against itself, this probably doesn't impact the main takeaways from your paper, but I think it is important to consider and an important question for the field: how do we validate the hazard estimates?

Figure 2 It would be helpful to see the univariate distributions of the two variables (rainfall and NTR). Here, you only show combinations that exceed either ~40 mm in 18-hr or ~0.6 m NTR. Why were these values selected to threshold the data? Given your findings that events with combined boundary conditions much, much smaller than the 1% AEP event can still generate flooding in excess of 1%. Do you have any concerns that there many be small events that should have been included in your stochastic event set?

Figure 2 What do your rainfall return period values look like compared to those estimated by NOAA Atlas 14? Is this distribution derived from the entire precipitation record (and over what scale)? I recognize that this information is likely provided in Maduwantha et al., but I think it is also important to reference that information in this paper as not every reader will have previously read Maduwantha.

Figure 2 How often is the NTR observed the Delaware River driven by coastal vs riverine flood events?

Figure 2 I would recommend splitting Figure 2 into two separate figures. One which shows panel a and a separate figure (and figure caption) for panels b-d. In addition, I would add a second column that shows the spatial distribution of the rainfall plotted next to each time series graph, similar to those shown in Supplementary Figures S5 and S6. This information is really valuable for understanding the resulting differences in flood inundation generated from these events that are described later in the paper.

Line 208-210 What is the size of the domain (total number of cells) in the model? Is 10 m the resolution of your model and 1 m the resolution of the subgrid?

Line 208 You state that the DEM was retrieved from CoNED. Does CoNED provide an accurate estimate for the channel bathymetry for the creeks in your model domain? Channel bathymetry is a big unknown in many hydrodynamic simulations and has been shown to be critical to correctly estimating flood inundation (e.g., they are too shallow, more water will be routed over the floodplain whereas if they are too deep, not enough water will be routed onto the floodplain). If they are not well-represented in CoNED, did you make any effort to 'burn in' or manually adjust the depth and width of the channels in your model? If not, to what extent do you think poor channel bathymetry may impact your validation results, particularly when you have to assume no infiltration in order to match a flood event in 2009 (lines 239-251)?

Line 237 You cite Flores et al. 2023 but there are other, more authoritative sources on this topic. Perhaps this study by the National Academies would be worth citing here in addition to Flores et al.

Lines 239-251 You do an excellent job of using what information you could find to try and validate the model. However, your primary focus is on infiltration and there is no discussion of whether local stormwater management (i.e., subsurface drainage) exists and whether that might contribute to error in the model performance since it can't be represented by SFINCS. There is a late reference to backwater (surcharges), but it does not come up in the validation. I would also suggest to include a description of the watershed as fully urbanized to help justify the decision to neglect infiltration.

Line 244-246 I would expect that neglecting infiltration has bigger implications for the smaller return period events and for those with low-intensity over the duration of the storm. Given that you are later comparing the design storms which only contain a few (or one?) event(s) with these characteristics against the probabilistic set which may contain many storms with these characteristics, what impact might

this have on your final results. How could antecedent conditions in the 2009 events contributed to your choice to neglect infiltration? Could the land use (urban?) in this area also provide a justification for neglecting infiltration?

Line 275 Given that SFINCS is very computationally inexpensive, another option for validation would have been to increase the size of the model domain to include locations with gages. For example, if you increase the model to the scale of the HUC10, it appears that the Cooper River watershed, adjacent to Newton Creek has two active USGS gages. Knowing how the model performs at these locations would give you confidence in the parameterization of your model (e.g., infiltration, roughness, elevation), even if your later simulations are focused on a subarea of the entire model domain.

Line 285 I find it interesting that there was little variability in the pluvial flood extent between the different 1% AEP events. How different were the simulated rainfall intensities for these storms or did they all have similar spatial-temporal distributions? Were their centers of mass all located in the same place? Perhaps a supplemental figure that provides the boundary conditions associated with the 25 events would be useful (similar to Figure 2b-d) but with the addition of a panel showing the spatial distribution of the rain that fell.

Line 377 Replace 'Lots' with 'Much'

Line 410 You state that ignoring the spatio-temporal variability of extreme events by relying on a design event can lead to significant uncertainties in flood exposure, but it does not appear that you prove this point. In both your cases (event- and response-based) you are selecting events that vary in space and time, you just sample them differently. This point sounds more like its drawing a comparison with the standard FEMA approach to selecting design storms that are uniform in space and vary in time according to a characteristic distribution. One way to address this would be to cite the existing literature when you make this point before discussing how your approach (both event- and response-based) provide more information than the standard design storm approach.

Line 455-462 You again write here that 'not accounting for the spatial variability of rainfall…' Perhaps I misread, but per the previous comment, I understood that for both your event and response-based approaches, the precipitation was spatially and temporally varied across the model domain. This raises an interesting question… to what extent are your findings driven by the spatial variations in the rainfall in the rainfall field vs the temporal variations in the rainfall distribution? Could one take a design storm approach in which you select the accumulated rainfall from the curve (Figure 2) and apply a characteristic distribution to distribute it over time, compare that to the situation in which you allow it to be temporally varying, and then against the full Monte Carlo approach to provide insight into whether it's the spatial variability or the temporal variability of the rainfall that really matters for the flood hazard? At a minimum, I would be more intentionally about what conclusions can be drawn from the findings in this analysis vs those that are made more broadly in the literature.

---

## Author Comment (AC1)

**Summary**

This study compares the flood hazard estimates generated using an event-based approach to those generated using a full suite of synthetic events, referred to herein as a response-based approach. Using Glouchester, NJ as a case study, the researchers estimate the compound hazard resulting from the joint probability distribution of 18-hr total precipitation and non-tidal residuals. From this distribution, they generate 5,000 synthetic storm events, each of which has a statistically derived spatiotemporally varying rainfall field and storm tide hydrograph (non-tidal residual + tide). From these, the authors sample 25 synthetic storm events with an Annual Expected Probability (AEP) of 1% (i.e., 'event-based') sampling from the isoline (representing maximum likelihood) and compare the estimated 1% AEP inundation map generated using the entire suite of synthetic storm events (i.e., 'response-based'). They find that the response-based approach generally generates higher hazard estimates, suggesting that selecting the 1% events based on boundary conditions alone does not generate the 1% hazard.

*The authors thank the Reviewer for the thoughtful and constructive feedback. Below, we provide detailed responses to each comment, along with explanations of how the suggested changes will be incorporated into the revised manuscript.*

Overall, I found this study very interesting to read and many of my comments and questions come from a place of genuine interest in the modelers' choices. My major review comments are as follows:

First, while the experimental design itself is not particularly novel, the combination of statistically-derived synthetic storm events that can represent compound drivers (both non-tidal residual and precipitation) as inputs to a process-based model takes many of the concepts that have been discussed within the compound hazard modeling community a step further than previous studies, and thus I find that it is worthy of publication.

*We thank the reviewer for the positive feedback on our approach and its contribution to advancing compound flood hazard modeling.*

Second, while this work is new in the context of compound flooding – a topic of considerable interest – the authors rely heavily on the compound flood literature. In doing so, they overlook a larger body of research regarding stochastic flood hazard simulation for rainfall-induced floods. For example, there have been several other studies that draw similar conclusions about the use of design storms vs stochastically generated (see, e.g., Perez et al. 2024, several papers by Daniel B. Wright). The paper could be further improved

by placing itself in this body of work, perhaps as part of both the introduction and discussion.

*We agree with the reviewer that the discussion of the research gap in the introduction and the interpretation of the results leaned heavily toward coastal drivers and compound events, reflecting the authors' stronger background in this area. We appreciate the reviewer's comment and the helpful references provided. In response, we have expanded the introduction and the discussion to better incorporate previous work on rainfall-induced flood hazards.*

*"Similarly, for rainfall and river discharge, traditional approaches defined a single "design storm" or "design event" to represent the temporal and spatial patterns of these drivers (i.e. a representative event structure). However, recent studies have shown that relying on a single "design storm", overlooking the variability in event structure across multiple storms, can underestimate flood hazards and associated impacts (Baer, 2025; Perez et al., 2024)."*

*"In contrast, the response-based approach can account for all these factors to produce more robust flood hazard estimates (Baer, 2025; Perez et al., 2024)."*

*"To our knowledge, the differences in flood hazard estimates between these two approaches have only been evaluated for rainfall flooding (Baer, 2025; Perez et al., 2024; Winter et al., 2020), but remain unexplored for compound coastal events. "*

*"These results are in line with previous studies that addressed the same question for rainfall-driven flooding (Baer, 2025; Perez et al., 2024; Winter et al., 2020)"*

*"Likewise, differences in rainfall-induced flooding between the event-based approach and the use of synthetic storms that capture the temporal and spatial variability of rainfall fields between events have been shown to significantly influence flood hazard estimates in the East and Gulf coasts of US (Baer, 2025; Perez et al., 2024) and Austria (Winter et al., 2020)."*

*"Methods such as the JPM, which expand the storm climatology, enable the generation of a larger set of extreme events (particularly tropical cyclones) and capture greater variability in their spatio-temporal characteristics compared to historical records. However, these methods are computationally demanding, as flood drivers must be generated in advance of the flood assessment using hydrologic and hydrodynamic models. Further research is needed to evaluate how different synthetic event generation approaches affect flood hazard estimates. Given the high computational demands of JPM, its application across large coastal areas may be impractical. One way to address this is optimal sampling, where a smaller set of events has to be run through hydrologic/hydrodynamic models while still capturing a large portion of the variability. Data-driven approaches like the one used in this study represent another efficient alternative. Similarly, other data-driven techniques, such*

*as stochastic storm transposition, are increasingly being adopted to generate synthetic rainfall fields for assessing rainfall-driven flood hazard (Baer, 2025; Perez et al., 2024; Winter et al., 2020). However, further investigation is needed to ensure that this method adequately preserves the interdependencies between coastal and rainfall processes when generating synthetic compound events for coastal flood assessments."*

Third, I commend the authors on their work to validate the hydrodynamic model in a data-scarce environment, however, I would like to see more discussion of the potential limitations of the synthetically generated boundary conditions. The proposed approach also introduces considerable uncertainties into the hazard estimates that are difficult to disentangle, including questions about how the authors are accounting for different flood types (e.g., those driven by high river events vs coastal storms) and whether the statistically-derived framework can adequately account for these.

*We thank the reviewer for this valuable comment. We agree that the manuscript's discussion of the limitations and uncertainties in the synthetic event generation approach was limited. While these issues are explored in more detail in a separate manuscript currently under review (Maduwantha et al., 2025), we recognize the importance of addressing them here, particularly in the context of flood hazard estimation.*

*Our data-driven framework is constrained by the length of the historical record (~100 years for point data and ~40 years for gridded precipitation), which limits the representation of rare or unprecedented compound events, especially those involving tropical cyclones. The stratification method used (based on a 350 km radius) may also lead to misclassification of events, further impacting the characterization of extremes due to small sample sizes. While this statistical framework accounts for dependencies of more key characteristics of the events (i.e., magnitude of the peaks, duration, time lags, intensities, etc.) than previous approaches, we acknowledge the difficulty in fully quantifying uncertainties given the stochastic nature of the drivers and potential long-term trends due to climate change. These uncertainties related to synthetic events potentially not capturing the full range of climate variability would affect the estimated flood hazards, but not the general findings and conclusions of the study. We validated the framework by comparing observed and simulated distributions of key event characteristics (i.e., magnitude of the peaks, duration, time lags, intensities, etc.) and dependencies among them, adding some constrains preventing the model from generating events that might not be physically possible (e.g., our maximum simulated peak NTR (2.9 m) and 18-hour rainfall (188 mm) are not much higher than the largest observed events in the records). One potential way to further validate the statistical framework's ability to represent low-probability events (primarily TCs) and to assess their impact on flood hazard estimates would be to compare its results with those*

*from an approach that uses the Joint Probability Method (JPM) to generate synthetic storms based on storm parameter distributions, followed by dynamic downscaling to the study site. Such a comparison would provide insight into the differences in driver variability captured by each method and their influence on compound flood hazard estimates. The authors intend to pursue this as part of future research.*

*Regarding the separation of coastal- vs. river-driven events, we agree with the reviewer that these can have different characteristics that are not specifically modeled. However, the location of our study site, within the tidal portion of the Delaware Estuary, makes such separation challenging due to the continuous influence of both drivers and their non-linear interactions. Disentangling and recombining these signals for synthetic event generation would be challenging and introduce significant uncertainties. Based on recent work (e.g., McKeon & Piecuch, 2025), we assume most NTR events at our site are primarily coastal in origin, though some include riverine influences. Our synthetic events are generated by sampling from observed NTR time series, which inherently reflect a mix of event types (although coastal dominated). While we do not explicitly classify events during generation, this historical variability allows us to include a range of compound event scenarios. Still, we acknowledge that the method may not fully capture unobserved combinations of coastal and fluvial drivers.*

*We have added the following lines into the manuscript to address this comment:*

"Another limitation of our study is that we use a synthetic event set developed using a data-driven statistical framework, which is limited to observed events. Although the statistical framework used to generate the synthetic events account for more dependencies between parameters that characterize the events (e.g. time lags) than other previous frameworks (Couasnon et al., 2018; Moftakhari et al., 2019a), it may not fully capture the full range of the potential spatio-temporal variability of flood drivers. Tropical cyclones might also be underrepresented in the historical sample since their frequency of occurrence is very low. This limitation can be overcome by using synthetic tropical cyclones that are dynamically downscaled to the study site (e.g., Gori et al., 2020). Methods such as the JPM, which expand the storm climatology, enable the generation of a larger set of tropical cyclones, and capture greater variability in their spatio-temporal characteristics compared to historical records. However, these methods are computationally demanding, as flood drivers must be generated in advance of the flood assessment using hydrodynamic models. Further research is needed to evaluate how different synthetic event generation approaches affect flood hazard estimates. Given the high computational demands of JPM, its application across large coastal areas may be impractical, making data-driven approaches like the one used in this study a more efficient alternative. Similarly, other

data-driven techniques, such as stochastic storm transposition, are increasingly being adopted to generate synthetic rainfall fields for assessing rainfall-driven flood hazards (Baer, 2025; Perez et al., 2024; Winter et al., 2020). However, further investigation is needed to ensure that this method adequately preserves the interdependencies between coastal and rainfall processes when generating synthetic compound events for coastal flood assessments."

*"Since our study site is located along the Delaware Estuary, the NTR reflects contributions from both fluvial discharge and coastal storm surge, as well as their nonlinear interactions. We opted not to disaggregate the NTR into riverine and coastal components due to the complexity of their coupled dynamics and the additional challenges and uncertainties this would introduce into the structure and parameterization of the multivariate statistical model."*

The comments below primarily ask for additional clarification or suggest that the authors contextualize some of their findings.

**Comments**

Line 34 the authors differentiate between event-based and response-based approaches. Is 'response-based' a commonly used term in flood hazard literature? Is there a citation that can be used to support this terminology? I have often heard them differentiated as 'design-storm' vs 'probabilistic' or 'stochastic' methods. Is there a citation that can be included to help support this choice of terminology?

*We appreciate the reviewer's observation regarding the varied terminology used in the literature. In the flood literature, different terms have been used for both the event and response-based approaches. Terms such as "design-storm" or "deterministic" have often been employed to describe what we refer to as the "event-based" approach. These terms typically involve estimating the probability of the flood drivers themselves, which can sometimes be misinterpreted as "probabilistic" methods. However, the term "probabilistic" has been applied across a range of methodologies: from those that consider the full probability distribution of flood drivers (Kupfer et al., 2024), to others that explore the uncertainty in flood model parameters for individual scenarios (e.g., Alfonso et al., 2016; Bates et al., 2004; Di Baldassarre et al., 2010).*

*We adopted the term "response-based" because of its use in the structural reliability literature, particularly in multivariate statistical contexts (see e.g. Gouldby et al., 2014; Jane et al., 2022), where the response variable (in our case, flooding) is influenced by multiple interdependent inputs, and different combinations of these inputs can yield similar*

*outcomes. However, upon further review of recent literature, we recognize that alternative terms such as "weather-generator-based" or "continuous simulation" approach (e.g. Winter et al., 2020) could have also been used instead of "response-based". We have added the following text in the introduction to acknowledge the different terminology used in the literature.*

*"There is no clear consensus in the literature regarding the terminology used to distinguish these two approaches. The event-based method is often referred to as the "design-storm" or "deterministic" approach. In contrast, the response-based approach has been described using terms such as "probabilistic," "stochastic," "continuous," or "weather-generator-based." However, some of these terms (particularly "probabilistic") are also used in other contexts, such as to describe flood maps that incorporate uncertainty in model parameters, which can lead to ambiguity in their interpretation (Alfonso et al., 2016; Bates et al., 2004; Di Baldassarre et al., 2010). Therefore, in this study, we adopt the term "response-based," consistent with its usage in the structural reliability literature (Gouldby et al., 2014; Jane et al., 2022)."*

In this same paragraph there is some discussion of the FEMA regulatory maps and process used for flood hazard delineation in the U.S. It is perhaps necessary here to differentiate from the approaches used for inland and coastal hazard. While they both rely on event-based approaches, the inland hazards are derived from design storms and assume a one-to-one relationship between the AEP of the precipitation event (spatially uniform, but varied in time according to a characteristic distribution) and the resulting flood flow which is then translated into hazard. In line 43, you state that temporal variability for inland flooding is neglected, but this is not true in the case of rainfall. It is true, however, in the case of flood mapping, where the peak flow is used to estimate the extent of the floodplain. There is some nuance here that gets lost in the way it is written now, and it might be worth revisiting this section.

*We thank the reviewer for highlighting once again our focus on coastal approaches while overlooking inland regions. In response, we have added the following text to include the approach used by FEMA for inland areas.*

*"For inland flooding, FEMA applies the event-based approach that starts by defining a design rainfall storm, typically derived from NOAA Atlas 14 which provides rainfall depths for specific probabilities and event durations (e.g., 1% AEP, 24-hour storms). The design storms are used in hydrologic models to simulate runoff, with the resulting hydrographs then routed through hydraulic models to estimate flood depths and extents."*

With respect to coastal flood hazard estimation, I'm not sure I agree with what is written in lines 43-49. Perhaps you are referring to how boundary conditions are applied to the downstream end of an inland model(?), but with respect to the FEMA SFHA (V-zone), it is my understanding that the most recent version is derived from ADCIRC simulations (where events are created using JP-OMS) in which case the spatial and temporal variability of the drivers and also the resulting water levels are considered when estimating the resultant hazard.

*The reviewer is correct that FEMA's most recent coastal flood mapping incorporates synthetic storm simulations using the JPM and ADCIRC; however, this approach has so far been applied only in some coastal regions. We have added the following text to address this comment:*

*"Similarly for coastal regions, a design event is selected from the distribution of coastal water levels to estimate the 1% AEP regulatory floodplain. In regions affected by tropical cyclones (TCs), FEMA further implements the Joint Probability Method (JPM) to construct synthetic storm climatology. This involves statistically sampling combinations of key storm parameters (e.g., central pressure deficit, radius to maximum winds, forward speed) based on their joint probability distributions. These synthetic events are then dynamically downscaled to the coast and exceedance probabilities of coastal water levels are calculated based on the probabilities of the storm characteristics. Although the JPM approach might reduce the uncertainties related to estimating the likelihood of low-probability coastal water level events, in both cases the probability of the event is assumed to approximate the probability of flooding."*

Line 57 remove 'up to a century'

*Removed*

Line 74-76 I think you are correct that this has not been done for compound flooding; however, I would point out two relevant studies that undertake a similar analysis applied to rainfall flooding (one that uses SFINCS) and reach the same conclusion. I think you may want to point to these both in your introduction (and draw differences between your work from theirs) and again in the conclusion.

Perez, G., Coon, E. T., Rathore, S. S., & Le, P. V. (2024). Advancing process-based flood frequency analysis for assessing flood hazard and population flood exposure. *Journal of Hydrology*, *639*, 131620.

Baer, J. A. (2025). *Design Storms Underestimate Flood Hazard and Risk Derived From Stochastic Storm Transposition* (Master's thesis, The University of North Carolina at Chapel Hill).

*As noted in our response to a previous, similar comment, we have expanded the introduction to incorporate the relevant studies highlighted by the reviewer. The following references have been added to the introduction:*

*"Similarly, for rainfall and river discharge, traditional approaches defined a single "design storm" or "design event" to represent the temporal and spatial patterns of these drivers (i.e. a representative event structure). However, some recent studies have shown that relying on a single 'design storm", overlooking the variability in event structure across multiple storms, can underestimate flood hazards and associated impacts (Baer, 2025; Perez et al., 2024)."*

*"In contrast, the response-based approach can account for all these factors to produce more robust flood hazard estimates (Baer, 2025; Perez et al., 2024)."*

*"To our knowledge, the differences in flood hazard estimates between these two approaches have only been evaluated for rainfall flooding (Baer, 2025; Perez et al., 2024; Winter et al., 2020), but remain unexplored for compound coastal events. "*

Line 88-90 What is the total size of the model domain in km2?

*The model domain is 147.45 $km^2$. We have added this information in the corresponding lines.*

Line 93 Provide date range for the start of the federal disaster declarations, e.g., 'Between X and 2016, there were five federal disaster declarations...'

*Added*

Line 94 Please provide more information about the building stock for context. You state that there are only 118 properties with coverage. Is this reflective of active policies based on OpenFEMA or where does this number come from? Do you also have the ability to provide a denominator for the total number of properties in Glouchester City and the number located within and outside of the FEMA SFHA? These would be useful numbers for context, given that insurance is only required at properties with a federally-backed mortgage within the SFHA.

*The information citing 118 properties with NFIP coverage was originally sourced from FEMA's Flood Risk Report for Camden County. To address the reviewer's request for clarification, we cross-checked the OpenFEMA dataset for "Individuals and Households Program – Valid Registrations" and found discrepancies. Specifically, for the year 2016 (the date of the Flood Risk Report) we identified only 94 properties with NFIP coverage, 76 of which are located within the Special Flood Hazard Area (SFHA). According to the National Structure Inventory, Gloucester City has 3,341 single-family homes, with 148 situated in the SFHA. We have updated the manuscript accordingly.*

*"Between 1974 and 2016, Gloucester City was subject to five federally declared flood-related disasters. Despite this, only 94 properties were enrolled in the National Flood Insurance Program (NFIP) as of 2016, according to data from OpenFEMA. Of these, 76 properties were located within the SFHA. Based on the National Structure Inventory, Gloucester City contains a total of 3,341 single-family homes, 148 of which are situated within the SFHA."*

Line 116-120 The downstream boundary condition at the Delaware River is assumed to be represented as a coastal water level? How do you consider the possibility of high river flow events (antecedent conditions that are not coastally driven) in your compound events framework? Have the previous works that are cited already established the relative frequency of high river flood events vs those that are driven by high coastal water levels and local rainfall? Do these events come from a different distribution than those that are considered 'compound coastal' events?

*The NTR, defined here as the "coastal boundary," is based on water level observations at the Philadelphia tide gauge and hence encompasses both river discharge and coastal storm surge, meaning that extreme NTR levels may result from either driver independently or from their combined influence. This integrated approach enables the modeling of both mechanisms while also capturing potential non-linear interactions between them, which is essential given our focus on compound flood events. Nonetheless, the dependence structure between NTR and rainfall may differ if the contributions from storm surge and river discharge are analyzed separately. Attempting to isolate and then recombine these components could introduce greater uncertainty in the resulting hydrographs due to the complexity of their interactions. Recent studies, such as McKeon & Piecuch (2025), which examined the relative contributions of coastal and fluvial influences in the Delaware Estuary, found that many events recorded at the Philadelphia tide gauge were predominantly driven by coastal processes (storm surge and tides). However, they also identified events driven solely by river discharge or by the interaction of both mechanisms. We have the following text to Section 3.1 (Synthetic compound events):*

*"Since our study site is located along the Delaware Estuary, the NTR reflects contributions from both fluvial discharge and coastal storm surge, as well as their nonlinear interactions. We opted not to disaggregate the NTR into riverine and coastal components due to the complexity of their coupled dynamics and the additional challenges and uncertainties this would introduce into the structure and parameterization of the multivariate statistical model."*

*And these lines in the discussion:*

*"A potential source of uncertainty in the variability captured by our synthetic event set arises from not disaggregating river- and coastal-driven components of the NTR. In our mid-estuarine study area, both processes contribute to the NTR, along with their nonlinear interactions. Separating these contributions would introduce considerable complexity due to their tightly coupled dynamics. Our approach is supported by recent work from (McKeon and Piecuch, 2025), who investigated the relative influence of coastal and fluvial drivers in the Delaware Estuary above flood thresholds. They found that most events observed at the Philadelphia tide gauge were primarily driven by coastal processes (e.g., tides and storm surge), but others resulted from river discharge alone or a combination of both mechanisms."*

Section 3.3 You invested considerable effort in validating the model, which I think was excellent given how limited the data for this cite is, however, the process used to generate the synthetic events for the model also has many layers of uncertainty in it (e.g., statistically derived joint probabilities of precipitation and non-tidal residual, statistically derived spatio-temporal rainfall structures, scaled event magnitudes, lag). To what extent could your model framework allow you to explore some of these uncertainties? Is there any mechanism that you could use to test the compound events framework and whether you believe your hazards estimates are true (assume the process-based model results are valid)? Given that you are comparing the model against itself, this probably doesn't impact the main takeaways from your paper, but I think it is important to consider and an important question for the field: how do we validate the hazard estimates?

*The authors agree with the reviewer that validating and quantifying the uncertainties related to synthetic storms/events is difficult due to the large natural variability of the stochastic climate and potential trends and changes induced by climate change. The authors have described and validated the data-driven statistical framework to generate the synthetic events against observed data to the best of our abilities in the following manuscripts (Maduwantha et al., 2024, 2025), having the second one under review. One of the options that the authors believe could help test the statistical framework for extreme events would be to compare the compound event hazard estimates of low-probability events, which are mostly TCs and the type of events of smaller sample size in the observed record and thus larger uncertainties, against synthetic TCs datasets dynamically downscaled to the study site. This would allow us to compare how well the statistical framework can capture their characteristics (e.g., return levels, spatio-temporal structures, lags...). We have extended the discussion about the uncertainties related to the synthetic compound event dataset.*

*We have added the following text to the section describing the methodology used for the synthetic event generation:*

*"The synthetic compound events were validated by comparing observed and simulated distributions of key event characteristics (e.g. magnitude of the peaks, duration, times lags, intensities) and dependencies among them, with good agreement between observed and simulated events."*

*And extended the discussion of the limitations and potential ideas for future research to address the point made by the reviewer:*

*"Another limitation of our study is that we use a synthetic event set developed using a data-driven statistical framework, which is limited to observed events. Although the statistical framework used to generate the synthetic events accounts for more dependencies between parameters that characterize the events (e.g. time lags) than other previous frameworks (Couasnon et al., 2018; Moftakhari et al., 2019a), it may not fully capture the full range of the potential spatio-temporal variability of flood drivers. Tropical cyclones might also be underrepresented in the historical sample since their frequency of occurrence is very low. This limitation can be overcome by using synthetic tropical cyclones that are dynamically downscaled to the study site (e.g., Gori et al., 2020). Methods such as the JPM, which expand the storm climatology, enable the generation of a larger set of tropical cyclones, and capture greater variability in their spatio-temporal characteristics compared to historical records. However, these methods are computationally demanding, as flood drivers must be generated in advance of the flood assessment using hydrodynamic models. Further research is needed to evaluate how different synthetic event generation approaches affect flood hazard estimates. Given the high computational demands of JPM, its application across large coastal areas may be impractical, making data-driven approaches like the one used in this study a more efficient alternative. Similarly, other data-driven techniques, such as stochastic storm transposition, are increasingly being adopted to generate synthetic rainfall fields for assessing rainfall-driven flood hazards (Baer, 2025; Perez et al., 2024; Winter et al., 2020). However, further investigation is needed to ensure that this method adequately preserves the interdependencies between coastal and rainfall processes when generating synthetic compound events for coastal flood assessments. A potential source of uncertainty in the variability captured by our synthetic event set arises from not disaggregating river- and coastal-driven components of the NTR. In our mid-estuarine study area, both processes contribute to the NTR, along with their nonlinear interactions. Separating these contributions would introduce considerable complexity due to their tightly coupled dynamics. Our approach is supported by recent work from (McKeon and Piecuch, 2025), who investigated the relative influence of coastal and fluvial drivers in the Delaware Estuary above flood thresholds. They found that most events observed at the Philadelphia tide*

*gauge were primarily driven by coastal processes (e.g., tides and storm surge), but others resulted from river discharge alone or a combination of both mechanisms.*

*Another limitation of the synthetic event set used is the reliance on mathematically defined thresholds for event selection, rather than thresholds based on actual flood impacts. This approach may exclude relatively frequent, lower-magnitude events that fall outside the statistical tails of the drivers' distributions but are still capable of causing localized flooding, potentially influencing response-based flood estimates. In our study, we evaluated the flood response of events near the selected thresholds and found that several produced no flooding, while others resulted in minor flooding, with empirical return periods between 1 and 2.8 years. As a result, the selected thresholds did not affect our response-based flood estimates; however, this may not hold true in other regions with different hydrologic or exposure characteristics."*

Figure 2 It would be helpful to see the univariate distributions of the two variables (rainfall and NTR). Here, you only show combinations that exceed either ~40 mm in 18-hr or ~0.6 m NTR. Why were these values selected to threshold the data? Given your findings that events with combined boundary conditions much, much smaller than the 1% AEP event can still generate flooding in excess of 1%. Do you have any concerns that there many be small events that should have been included in your stochastic event set?

*We agree with the reviewer that the threshold should be selected depending on the application, and in terms of flooding, the threshold should be selected based on an elevation (or rainfall rate) that starts flooding. However, the multivariate statistical models used are based on the statistical distribution of the variables, and thus the threshold should define the tail of the distribution. However, selecting a threshold is not a straightforward task, and subjectivity is included. We have selected a threshold that balances the events' sample size (large enough to reduce the uncertainties) with the magnitude of the drivers (to capture the tail of the distribution). We decided to set the threshold to obtain an average of 5 events per year from each of the drivers. We performed a sensitivity analysis of the dependence between the drivers, finding that selecting an average of 5 events per year also gave us the strongest dependency between the drivers. However, the two-way conditioning sampling allows the selection of events of smaller magnitude than the threshold for the other driver. This also allows to account for compound events in which one of the drivers might not be extreme. Still, since these events are selected following mathematical conditions instead of physical processes (i.e. flooding), we have checked the flooding arising from the events with the largest joint probabilities (smallest return periods), finding that the events closest to both the rainfall and NTR thresholds cause none or very little flooding, with response-based return periods between 1 and 2.8 years. Therefore, including*

*smaller events might affect the response-based estimates on the lower part of the tail, but their effect on the 1% AEP flood would be negligible.*

*We have added the following lines to include details about threshold selection*

*"Observed compound events were identified using the Peaks Over Threshold (POT) approach combined with a two-sided conditional sampling method. Thresholds were set to capture an average of five events per year, providing a balance between sufficient sample size and an appropriate representation of the distribution tail. These thresholds also maximized the statistical dependence between variables. Additionally, the conditional sampling method includes events where one variable is not extreme, allowing for coverage of the full range of driver magnitudes, including those that may not lead to flooding. "*

"Another limitation of the synthetic event set used is the reliance on mathematically defined thresholds for event selection, rather than thresholds based on actual flood impacts. This approach may exclude relatively frequent, lower-magnitude events that fall outside the statistical tails of the drivers' distributions but are still capable of causing localized flooding, potentially influencing response-based flood estimates. In our study, we evaluated the flood response of events near the selected thresholds and found that several produced no flooding, while others resulted in only minor inundation, with empirical return periods between 1 and 2.8 years. As a result, the selected thresholds did not affect our response-based flood estimates; however, this may not hold true in other regions with different hydrologic or exposure characteristics."

Figure 2 What do your rainfall return period values look like compared to those estimated by NOAA Atlas 14? Is this distribution derived from the entire precipitation record (and over what scale)? I recognize that this information is likely provided in Maduwantha et al., but I think it is also important to reference that information in this paper as not every reader will have previously read Maduwantha.

*NOAA Atlas 14 estimates for 18-hour rainfall in this region are approximately 163 mm and 187 mm for the 50-year and 100-year return periods, respectively (based on linear interpolation between the published 12-hour and 24-hour durations). However, these values are not directly comparable to the 18-hour univariate rainfall return levels produced by our statistical framework. This is because NOAA Atlas 14 provides "point rainfall estimates" while our framework estimates "basin-averaged rainfall values", representing the spatially averaged precipitation over the entire catchment area.*

*Our 50-year and 100-year 18-hour basin-average rainfall return levels are 124 mm and 139 mm, respectively. These are lower than the corresponding point estimates, as expected, since spatial averaging smooths localized peaks. When scaling a historical rainfall field to*

*match a synthetic rainfall peak, the observed basin-average peak is called to match the synthetic peak. However, individual grid cells within the basin may exhibit values higher or lower than the basin average, reflecting the spatial heterogeneity in the original historical rainfall field.*

*The rainfall (joint probability) distribution is derived using the rainfall data of Philadelphia airport from 1900 to 2021 and AORC (Analysis of Period of Record for Calibration) data from 1979 to 2021 over the selected catchment. We apply a bias correction to the hourly RF gauge data to match the hourly basin-average RF values calculated from AORC. The bias correction is performed using the quantile mapping method, fitting both the hourly measured gauge data and the hourly AORC basin-average data to gamma distributions.*

Figure 2 How often is the NTR observed the Delaware River driven by coastal vs riverine flood events?

*As previously noted, we have not separated or analyzed the NTR (non-tidal residual) events based on coastal and riverine contributions due to the complex interactions between these components. According to the analysis by (McKeon & Piecuch, 2025), most minor flood events at the Philadelphia tide-gauge (defined using NOAA's minor flooding thresholds) are primarily driven by coastal drivers (storm surge and tides). However, some events are mainly attributed to river discharge combined with tides, or to a mix of all three factors. They also report that the "residuals" (water level variations not accounted for by open coast surge, tides, or upstream river discharge) are substantial at this site, likely due to non-linear interactions among these contributing elements. Therefore, we assume that the NTR events we identified are primarily driven by coastal processes, although some may also be influenced by river discharge and non-linear interactions among the contributing drivers.*

Figure 2 I would recommend splitting Figure 2 into two separate figures. One which shows panel a and a separate figure (and figure caption) for panels b-d. In addition, I would add a second column that shows the spatial distribution of the rainfall plotted next to each time series graph, similar to those shown in Supplementary Figures S5 and S6. This information is really valuable for understanding the resulting differences in flood inundation generated from these events that are described later in the paper.

We recognize the added value of including the rainfall fields; however, splitting the figure into two would make it more difficult to visualize the location of events within the joint probability space. Similarly, incorporating the rainfall fields directly into the figure would result in an excessively large and cluttered layout. The main flood differences among these events occur along the city's waterfront and are primarily driven by variations in water levels. In response to another reviewer's comment, we have included flood hazard maps

for all events with a 1% AEP in the Supporting Material. Therefore, we believe that splitting Figure 2 is no longer necessary and have chosen to retain its current format.Line 208-210 What is the size of the domain (total number of cells) in the model? Is 10 m the resolution of your model and 1 m the resolution of the subgrid?

*Yes, the resolution of the model is 10m and the subgrid is 1m. At 10 m, the model has 758,904 cells and at 1m it has 147,450,698 cells. We have added this information to the manuscript.*

Line 208 You state that the DEM was retrieved from CoNED. Does CoNED provide an accurate estimate for the channel bathymetry for the creeks in your model domain? Channel bathymetry is a big unknown in many hydrodynamic simulations and has been shown to be critical to correctly estimating flood inundation (e.g., they are too shallow, more water will be routed over the floodplain whereas if they are too deep, not enough water will be routed onto the floodplain). If they are not well-represented in CoNED, did you make any effort to 'burn in' or manually adjust the depth and width of the channels in your model? If not, to what extent do you think poor channel bathymetry may impact your validation results, particularly when you have to assume no infiltration in order to match a flood event in 2009 (lines 239-251)?

*CoNED is a lidar-derived topobathymetric dataset with a 1-meter resolution, and to our knowledge, it is the most current and highest-resolution dataset available for our study area. However, we were unable to find information on its accuracy in representing creek channels within our site, nor did we find any other DEMs that provide detailed information about these features or had the opportunity to collect the data ourselves. Therefore, we have not modified the bathymetry of these channels. We could only revise satellite images and street-view photos to better understand the potential depth of these channels. These sources suggest that the channels may be a few meters deep near their intersections with the Delaware River, but become much shallower and more vegetated further inland. We have included a map of the topobathymetric dataset used in the Supporting Material. As can be seen in it, the channels are represented in the DEM, though their depth may be underestimated. We have addressed this limitation, as well as the uncertainty in water depth levels around the creeks due to the exclusion of infiltration, adding:*

*"Although most of our study domain is urban and thus covered by impervious surfaces, we might underestimate infiltration in areas with larger amounts of vegetation such as areas around creeks. The bathymetry of the creeks might also not be accurately represented in the CoNED DEM (Fig. S1). If that was the case, combined with the exclusion of infiltration processes, our approach could overestimate floodwater depths along the margins of the creeks. "*

[Figure]

**Figure S1 Topobathymetry from CoNED (NAD83/UTM18N)**

*We do not believe that the validation of the 2009 event is affected by inaccuracies in the creeks' channels since the location of the photo used for validation is in an area far from the creeks and not hydrologically connected to the creeks (at least for an event of that magnitude and that was rainfall-driven).*

Line 237 You cite Flores et al. 2023 but there are other, more authoritative sources on this topic. Perhaps this study by the National Academies would be worth citing here in addition to Flores et al.

*We appreciate the reviewer's suggestion, and we have added the study.*

Lines 239-251 You do an excellent job of using what information you could find to try and validate the model. However, your primary focus is on infiltration and there is no discussion of whether local stormwater management (i.e., subsurface drainage) exists and whether that might contribute to error in the model performance since it can't be represented by SFINCS. There is a late reference to backwater (surcharges), but it does not come up in the validation. I would also suggest to include a description of the watershed as fully urbanized to help justify the decision to neglect infiltration.

*We thank the reviewer for highlighting this important issue. In response, we have expanded the description of land cover in the study area and included a land cover map in the Supporting Information. We also address the challenges faced by municipalities within the catchments due to inadequate stormwater infrastructure, particularly the use of combined sewer systems. Additionally, we have included a note on the initial soil moisture assumption in the Curve Number method implemented in SFINCS, which presumes soils are at 50% of their total capacity, a simplification that has been shown to potentially overestimate infiltration (e.g., Nederhoff et al., 2024). These considerations have also been incorporated into the discussion of the model validation and the study's limitations.*

*"This, combined with the highly urbanized nature of the catchment (characterized by extensive impervious surfaces) and the inadequate stormwater system performance reported by the CCMUA, led us to exclude infiltration in the SFINCS model configuration used for all simulations in this study. At the time this study was conducted, the Curve Number method was the most advanced infiltration approach available in SFINCS (the latest release has since added the Green-Ampt method). However, as noted by Nederhoff et al. (2024), assuming that the initial soil moisture is at 50% of its total capacity can lead to an overestimation of infiltration."*

[Figure]

**Figure S2 Land Cover of the study site. Note that the land cover classes have been grouped in these main classes for plotting purposes, but the original dataset contains a larger number of classes (NAD83/UTM18N)**

*"Nevertheless, the exclusion of stormwater infrastructure may have a greater impact on the results for smaller, more frequent events, potentially leading to an overestimation of flooding in cases where the existing drainage system would likely manage the runoff. However, this would have a negligible impact on the response-based estimates for the 1% AEP since the empirical distribution will not change for rare large events. "*

Line 244-246 I would expect that neglecting infiltration has bigger implications for the smaller return period events and for those with low-intensity over the duration of the storm. Given that you are later comparing the design storms which only contain a few (or one?) event(s) with these characteristics against the probabilistic set which may contain many storms with these characteristics, what impact might this have on your final results. How could antecedent conditions in the 2009 events contributed to your choice to neglect infiltration? Could the land use (urban?) in this area also provide a justification for neglecting infiltration?

*Neglecting infiltration will likely produce an overestimation of the flood results, which will be more noticeable for small return period events and in vegetated areas. Since we primarily focus on events of 1% joint probabilities, we assume that these effects are small/negligible and will not affect the main conclusions of the analysis. The response flood hazard is estimated based on the empirical maximum water depth distribution at each model cell, and thus, reducing the water depth corresponding to small events will not have an impact on the 1% water depth (as discussed previously).*

*In line with our response to the previous comment and the findings of Nederhoff et al. (2024), we consider that one of the primary factors contributing to the significant overestimation of infiltration is the fixed initial condition assuming the soil is at 50% of its storage capacity. At the time this study was conducted, this parameter could not be modified within the model. As the reviewer correctly notes, antecedent conditions (such as those preceding the 2009 event) can influence actual soil moisture levels, potentially making the 50% assumption inaccurate. We reviewed rainfall data from the two days preceding the 2009 event and found no recorded precipitation, suggesting that the soil was likely dry. We believe that the curve number method overestimates infiltration for this event, particularly in areas dominated by urban impervious surfaces.*

Line 275 Given that SFINCS is very computationally inexpensive, another option for validation would have been to increase the size of the model domain to include locations with gages. For example, if you increase the model to the scale of the HUC10, it appears that the Cooper River watershed, adjacent to Newton Creek has two active USGS gages. Knowing how the model performs at these locations would give you confidence in the parameterization of your model (e.g., infiltration, roughness, elevation), even if your later simulations are focused on a subarea of the entire model domain.

*We acknowledge the reviewer's suggestion that extending the model domain to include gauged locations and using gauge records for validation is a well-established and widely used practice, and we considered this approach during the study. However, we concluded that it would not significantly improve the validation of our flood model for the specific study area, as the dominant physical processes in riverine systems differ from those governing urban overland flooding. For instance, infiltration dynamics in river environments contrast sharply with those in densely urbanized regions. Surface roughness also varies considerably between riverbeds and urban surfaces. Furthermore, bathymetric uncertainties in the main river (affected by factors such as water color and vegetation cover, which can influence lidar accuracy) are not directly comparable to those in the smaller creeks of our domain. Considering these factors, along with the high computational*

*cost of extending the high-resolution model domain and the large number of simulations required, we decided not to implement this type of validation.*

Line 285 I find it interesting that there was little variability in the pluvial flood extent between the different 1% AEP events. How different were the simulated rainfall intensities for these storms or did they all have similar spatial-temporal distributions? Were their centers of mass all located in the same place? Perhaps a supplemental figure that provides the boundary conditions associated with the 25 events would be useful (similar to Figure 2b-d) but with the addition of a panel showing the spatial distribution of the rain that fell.

*There is substantial variability in the location and extent of pluvial flooding hotspots across the different 1% AEP events, with accumulated catchment-average rainfall ranging from 23.26 mm to 132 mm. However, water depths in these areas are generally much lower than those in coastal zones, where the interaction of rainfall and coastal drivers results in significantly greater combined variability. Since both pluvial and coastal flooding are displayed on the same maps, we used a consistent color scale, which limits the visibility of variation in the pluvial hotspots. Although we tested several colormaps and scaling options, we were unable to meaningfully enhance the representation of this variability. To address this, we included the 25 flood maps corresponding to the 1% AEP events in the Supporting Material to better illustrate the spatial variability in pluvial flooding. However, we opted not to include the spatial and temporal distributions of the rainfall events, as doing so would require 50 additional figures and would not effectively convey the differences in flooding outcomes for these locations.*

Line 377 Replace 'Lots' with 'Much'

*Done*

Line 410 You state that ignoring the spatio-temporal variability of extreme events by relying on a design event can lead to significant uncertainties in flood exposure, but it does not appear that you prove this point. In both your cases (event- and response-based) you are selecting events that vary in space and time, you just sample them differently. This point sounds more like its drawing a comparison with the standard FEMA approach to selecting design storms that are uniform in space and vary in time according to a characteristic distribution. One way to address this would be to cite the existing literature when you make this point before discussing how your approach (both event- and response-based) provide more information than the standard design storm approach.

*The authors agree with the reviewer that it is challenging to isolate the effects of spatial and temporal variability on flooding, given that the sampled events vary in both space and time. As a result, it is difficult to distinguish whether differences in flooding outcomes are driven*

*by temporal variability or by spatial differences in the rainfall fields and/or the time lag between the peaks of the events. However, we do observe a clear example illustrating these effects: two events with nearly identical magnitudes in the joint probability space (i.e., both classified as 1% AEP meaning they have essentially the same NTR and rainfall total) exhibit significantly different spatial and temporal characteristics, resulting in notably different flood extents and water depths (among the largest observed across all 1% AEP events). If both events were simulated using the same design rainfall field and coastal hydrograph (i.e., with identical spatial and temporal characteristics and time lag), they would be expected to produce almost equal flooding outcomes. Additionally, we find that events with relatively high AEPs (i.e., lower magnitude) can still produce 1% AEP water depths when they have prolonged hydrographs with sustained high water levels and spatial rainfall distributions concentrated over Gloucester City. This indicates that event structure (both in time and space) plays a critical role in driving flood impacts, regardless of the overall event magnitude.*

*We also agree with the reviewer that the term "temporal and spatial variability" we used to describe the variations of temporal and spatial patterns between events can be misleading, since it can also refer to differences between events that are constant in time and space to those with varying temporal distributions and spatial fields. Therefore, we have rephrased those lines.*

*"Event-based flood assessments commonly use a single design event with specific temporal and spatial structure, thus neglecting the variability in the temporal and spatial evolution of the flood drivers between different events. We have shown that using a single 1% AEP design event can introduce large uncertainties in both flood extents and water depths that arise from the different combinations of the drivers' magnitude but are mostly due to differences in temporal and spatial evolution of events. Events of almost equal magnitude but different spatial rainfall fields and temporal distribution of the water level hydrographs can produce very different flood extents and water depths."*

Line 455-462 You again write here that 'not accounting for the spatial variability of rainfall...' Perhaps I misread, but per the previous comment, I understood that for both your event and response-based approaches, the precipitation was spatially and temporally varied across the model domain. This raises an interesting question... to what extent are your findings driven by the spatial variations in the rainfall in the rainfall field vs the temporal variations in the rainfall distribution? Could one take a design storm approach in which you select the accumulated rainfall from the curve (Figure 2) and apply a characteristic distribution to distribute it over time, compare that to the situation in which you allow it to be temporally varying, and then against the full Monte Carlo approach to provide insight

into whether it's the spatial variability or the temporal variability of the rainfall that really matters for the flood hazard? At a minimum, I would be more intentionally about what conclusions can be drawn from the findings in this analysis vs those that are made more broadly in the literature.

*The authors agree with the reviewer that it is not possible to isolate the individual effects of temporal and spatial variability in rainfall fields, as both vary simultaneously across events. Initially, we explored using a set of design storms with varying temporal and spatial structures (and time lags), rescaled to different points along the 1% AEP isoline. However, this approach can lead to unrealistic compound scenarios by combining spatial and temporal features that may not be physically plausible and are not easily validated. To provide a more robust and representative analysis of the temporal and spatial characteristics of compound events and their impact on flooding in the study area, we chose to use events generated by the data-driven statistical framework. This method captures the observed variability in event characteristics and their associated probabilities. While this limits our ability to attribute flood impacts to specific factors (e.g., temporal pattern, spatial distribution, or time lag), it offers a more realistic assessment of the consequences of neglecting inter-event variability. Finally, by "temporal and spatial variability," we do not imply that traditional or design-event approaches assume uniform time series or rainfall fields, but rather that they use a fixed temporal distribution and spatial pattern, thus omitting the broader variability observed in real events. Following our response to the previous comment, we have modified these lines to clearly reflect the previous point.*

*"Not fully accounting for the variability in the spatio-temporal structure between extreme events by relying on a single design event can lead to significant uncertainties in flood exposure, which in turn can result in substantial uncertainties in flood risks. One way to address this limitation when using the event-based approach is to employ ensembles of events where the flood drivers exhibit different spatial and temporal variability"*

---

## Author Comment (AC2)

**REVIEW 2**

This paper integrates synthetic storm generation (rainfall and storm surge) with a reduced-complexity hydrodynamic model to assess the impact of between-storm variability in flood hazard estimates. The authors find that using a single nominal design storm can provide a very limited understanding of flood hazard or risk, suggesting important directions for future work.

Overall, this paper makes a valuable and substantial contribution to the field of coastal flood risk assessment by highlighting the limitations of traditional design storm approaches and advocating for the use of large ensembles and probabilistic methods. However, the paper needs significant revisions to clarify its contributions and limitations, and to improve its readability.

*The authors would like to thank the Reviewer for the valuable and insightful comments. We address each point in detail below and describe how the corresponding suggestions will be implemented or considered in the revised manuscript.*

My primary concern is that the paper "buries the lede" by spending a lot of time on the specific test case and model development, which detracts from the main contribution of the work. The key findings about the limitations of design storm approaches and the benefits of large ensembles and probabilistic assessments are somewhat obscured by the detailed discussion of the modeling setup and validation. Fundamentally, I don't think the conclusions of the paper are particularly sensitive to specific details about how well-validated the SFINCS model for this watershed is (although there are of course many questions for which that would be important!) I'd suggest moving some of the testbed results and figures to an appendix or supplementary material.

*We appreciate the reviewer's comment. While we initially considered including the flood model validation in the Supporting Material, the authors view model validation as a critical component of any modeling study and have a critical view on studies that provide limited information about validation or insufficient detail. We also aim to include as much information as possible about the model setup to ensure transparency and reproducibility. That said, given the manuscript's primary focus on advancing flood hazard modeling and in line with the reviewer's recommendation, we have moved the validation section to the Supporting Material to better emphasize the core analyses and findings in the main text.*

A related concern is about the level of polish of the figures. Improving the clarity and presentation of the figures would greatly enhance the overall readability and impact of the paper. Just to pick the first figure as an example, fig 1 spends a large amount of space on the x and y axis labels, wasting space, is very low-resolution, lacks (a) (b) (c) markings, and the use of satellite imagery in the larger figure could more usefully be topography or land use. Similar critique applies to most of the figures in the paper. For another example, I found the number of different vertical lines with different colors and line styles in fig 3 to be confusing and distracting.

*We thank the reviewer for this helpful comment and have revised the figures to enhance their readability. Specifically, we updated Figure 1 to make more efficient use of space by reducing the font size of axes and labels to the minimum permitted by the journal and replacing the original maps with higher-resolution versions. In response to feedback from another reviewer, we have also added a map of the topobathymetric and land cover data used in this study to the Supporting Material. We chose to retain the satellite basemap in Figure 1, as we believe it offers valuable context about the study area. For the remaining figures, we reduced the font sizes of axes and labels to maximize the area available for data visualization. Additionally, in Figure 3, we replaced the lines representing tidal levels with triangle markers to reduce visual clutter while preserving key information.*

[Figure]

A third concern is about the discussion of probabilities in this paper, which I similarly find to be confusing and distracting to the reader. As an example, figure 6 talks about the "most likely 100 year event." This is particularly confusing, as a standard interpretation of the "100 year event" in a bivariate context is that there is a 1% chance of a draw from the bivariate distribution landing "outside the curve". While some definition is provided, it is confusing and I don't think that it necessarily supports the development of the paper. Section 3 says "We investigate differences in flooding between the event- and response-based

approaches by simulating flooding from a large number (5,000) of compound events that allow estimating the empirical distribution of flooding and comprise several 1% AEP events." This discussion is tortured. Although I understand what you're saying, I would suggest being more specific and explicit with your language. For example, instead of calling something a 25-year compound event or 4% AEP event, I would take a little bit more space and describe it as "an event whose rainfall or storm surge would be expected to be exceeded once every 25 years," which is a bit of a longer text but will greatly enhance the readability of the final work.

*We appreciate the reviewer's observation regarding the clarity of terminology. To improve this, we have added a brief definition of the 1% Annual Exceedance Probability (AEP) or 100-year event in the introduction. Throughout the manuscript, we have revised several instances of "1% AEP" to more intuitive phrasings such as "event with a 1% chance of occurring in any given year," particularly in the lines noted by the reviewer. However, we retained the terms "1% AEP" and "100-year event" in other parts of the text, as they are widely recognized in the field and help maintain conciseness in more technical or complex sentences.*

*Regarding the use of the term "most likely" event, we agree it can be misleading. However, as this terminology is commonly adopted in multivariate design frameworks to identify a "design event" (i.e., representative event) along an isoline based on joint probability density, we have chosen to retain it to ensure consistency with existing literature and to facilitate comparisons with standard event-based approaches in multivariate analyses ( see e.g., Gori et al., 2020; Jane et al., 2022; Moftakhari et al., 2019). Nevertheless, we have extended the explanations related to this approach in the introduction, methods and discussion sections.*

*The following text has been modified and added in the manuscript:*

*"For the latter, selecting a single 1% AEP design event is particularly challenging, as multiple combinations of flood drivers can yield the same joint exceedance probability. This challenge has sometimes been addressed through the use of ambiguous constructs, such as the "most likely" event, which attempts to identify a representative scenario among equally probable combinations based on the density of observed events (Jane et al., 2022; Moftakhari et al., 2019b; Salvadori et al., 2011a)."*

*"To further investigate differences in flood hazard estimations between approaches, we also define a "design event" from all the 100-year events following the "most likely" approach for multivariate events (Jane et al., 2022; Moftakhari et al., 2019a). This approach selects one event in the isoline based on the density of observed events along it (Salvadori*

*et al., 2011b), identifying this event as the most representative scenario ("most likely")
among the equally probable combinations along the isoline."*

My last concern is about the methodology used to generate the synthetic storms in a
probabilistic framework. The authors provide a brief overview and largely refer to a
previously published paper. It would be helpful to have a bit more detail here. In particular,
the ways in which the authors sample from this bivariate distribution, and how it
conceptually relates to other related approaches such as JPM-OS, climate model
downscaling, synthetic storm generation, etc. are not clear (to be clear; those methods
solve problems different from what the authors are doing here, so I am not suggesting that
they need to defend and justify their choice so much as explain conceptual links and
differences). The caveats and limitations of this approach ought to be more clearly
discussed in the discussion section, in particular the discussion of how the assumption of
uniform rainfall affects the finding that pluvial flooding is relatively insensitive to between-
storm variability, which is at odds with other previously published work.

*In response to this comment, along with related feedback from the other reviewer, we have
expanded the description of the multivariate statistical framework used to generate the
synthetic events. This includes additional details on the selection of extreme compound
events, the definition of NTR, and the validation procedures. We have also extended the
discussion of methodological limitations and clarified how this approach compares with
alternative methods, as suggested by the reviewer.*

*The following text has been included:*

*Introduction:*

*"For inland flooding, FEMA applies the event-based approach that starts by defining a
design rainfall storm, typically derived from NOAA Atlas 14 which provides rainfall depths
for specific probabilities and durations (e.g., 1% AEP, 24-hour storms). The design storms
are used in hydrologic models to simulate runoff, with the resulting hydrographs then
routed through hydraulic models to estimate flood depths and extents. Similarly for coastal
regions, a design event is selected from the distribution of coastal water levels to estimate
the 1% AEP regulatory floodplain. In regions affected by tropical cyclones (TCs), FEMA
further implements the Joint Probability Method (JPM) to construct synthetic storm
climatology. This involves statistically sampling combinations of key storm parameters (e.g.
central pressure deficit, radius to maximum winds, forward speed) based on their joint
probability distributions. These synthetic events are then dynamically downscaled to the
coast and exceedance probabilities of coastal water levels are calculated based on the*

*probabilities of the storm characteristics. Although the JMP approach might reduce the uncertainties related to estimating the likelihood of low-probability coastal water level events by increasing the sample size of this events, in both cases, the probability of the event is assumed to approximate the probability of flooding."*

*"Similarly, for rainfall and river discharge, traditional approaches defined a single "design storm" or "design event" to represent the temporal and spatial patterns of these drivers (i.e., a representative event structure). However, some recent studies have shown that relying on a single "design storm", overlooking the variability in event structure across multiple storms, can underestimate flood hazards and associated impacts (Baer, 2025; Perez et al., 2024) ."*

*"To our knowledge, the differences in flood hazard estimates between these two approaches have only been evaluated for rainfall flooding (Baer, 2025; Perez et al., 2024; Winter et al., 2020)(Baer, 2025; Perez et al., 2024; Winter et al., 2020), but remain unexplored for compound coastal flooding. "*

*Discussion:*

*"Likewise, differences in rainfall-induced flooding between the event-based approach and the use of synthetic storms that capture the breadth of temporal and spatial variability of rainfall fields have been shown to significantly influence flood hazard estimates in the U.S. East and Gulf coast regions (Baer, 2025; Perez et al., 2024)  and in Austria (Winter et al., 2020) , among others.*

*Another limitation of our study is that we use a synthetic event set developed using a data-driven statistical framework, which is limited to observed events. Although the statistical framework used to generate the synthetic events accounts for more dependencies between parameters that characterize the events (e.g. time lags) than other previous frameworks (Couasnon et al., 2018; Moftakhari et al., 2019a), it may not fully capture the full range of the potential spatio-temporal variability of flood drivers. Tropical cyclones might also be underrepresented in the historical sample since their frequency of occurrence is very low. This limitation can be overcome by using synthetic tropical cyclones that are dynamically downscaled to the study site . (e.g., Gori et al., 2020). Methods such as the JPM, which expand the storm climatology, enable the generation of a larger set of tropical cyclones, and capture greater variability in their spatio-temporal characteristics compared to historical records. However, these methods are computationally demanding, as flood drivers must be generated in advance of the flood assessment using hydrodynamic models. Further research is needed to evaluate how different synthetic event generation approaches affect flood hazard estimates. Given the*

*high computational demands of JPM, its application across large coastal areas may be impractical, making data-driven approaches like the one used in this study a more efficient alternative. Similarly, other data-driven techniques, such as stochastic storm transposition, are increasingly being adopted to generate synthetic rainfall fields for assessing rainfall-driven flood hazards (Baer, 2025; Perez et al., 2024; Winter et al., 2020). However, further investigation is needed to ensure that this method adequately preserves the interdependencies between coastal and rainfall processes when generating synthetic compound events for coastal flood assessments. A potential source of uncertainty in the variability captured by our synthetic event set arises from not disaggregating river- and coastal-driven components of the NTR. In our mid-estuarine study area, both processes contribute to the NTR, along with their nonlinear interactions. Separating these contributions would introduce considerable complexity due to their tightly coupled dynamics. Our approach is supported by recent work from McKeon & Piecuch (2025), who investigated the relative influence of coastal and fluvial drivers in the Delaware Estuary above flood thresholds. They found that most events observed at the Philadelphia tide gauge were primarily driven by coastal processes (e.g., tides and storm surge), but others resulted from river discharge alone or a combination of both mechanisms. Another limitation of the synthetic event set used is the reliance on mathematically defined thresholds for event selection, rather than thresholds based on actual flood impacts. This approach may exclude relatively frequent, lower-magnitude events that fall outside the statistical tails of the drivers' distributions but are still capable of causing localized flooding, potentially influencing response-based flood estimates. In our study, we evaluated the flood response of events near the selected thresholds and found that several produced no flooding, while others resulted in only minor inundation, with empirical return periods between 1 and 2.8 years. As a result, the selected thresholds did not affect our response-based flood estimates; however, this may not hold true in other regions with different hydrologic or exposure characteristics."*

*Conclusions:*

*"Additionally, future research should aim to evaluate how different methods for generating synthetic events influence the resulting flood hazard estimates. Such comparisons can help inform best practices for generating more reliable flood hazard assessments under both current and future climate conditions."*

Minor Points

I have a few specific suggestions that illustrate the above points, though they do not provide comprehensive wordsmithing for the paper.

- page 1: avoid "hinterland" word choice

*Changed to "inland areas"*

- page 5: Maduwantha.....characteristics: this seems like a lot of work to go through to avoid generating synthetic, realistic events

*We are a bit unclear on what exactly the reviewer meant here. We believe (and show through different validation steps) that our synthetic events are "realistic". As discussed in previous comments and our responses, there are other ways of generating synthetic events. If the reveiwer refers to the approach that generates synthetic storms, that approach is actually a lot more work to go through than what we use here, because a storm surge model has to be used with the wind and pressure fields, a hydrologic model with the rainfall fields, and a hydrodynamic model to link the two. While SFINCS can do many of those things it hasn't been tested (much) for it's ability to generate and propgate storm surge from the open ocean to the coast. It would also require to cover a very large domain (in our case essentially the entire Delaware basin) making it computationally very demanding as opposed to our approach. - page 11: "We find that the floodplain of each of these 1% AEP events is different, resulting in very large differences in both flood extent and depth between some of the events." this is perhaps an even better example of how the discussion of return periods here is confusing. What is an AEP event? I eventually was able to figure this out. A key finding of the paper is that there is not a 1:1 link between the return period of the rainfall rate or the storm surge and the return period of the flooding or the damage, in line with other studies. Calling something a 1% AEP event is not helpful here. I don't think the methods need to change, just the presentation.*

*Following the response to the previous comment, we have defined what 1% AEP means in the introduction and modified the use of 1% AEP for "event with a 1% chance of occurring in any given year" in several sentences along the manuscript. We have also modified the objective of the analysis in the introduction: "Here, we explore the degree of linearity in the relationship between events with 1% chance of occurring any year and flooding of equal probability, from compound events of precipitation and estuarine water levels in a case study for Gloucester City, New Jersey"*

*And in the discussion:*

*"Here, we have assessed the relationship between the probability of the event and the probability of flooding for a case study in Gloucester City (NJ, U.S.) by comparing the flood hazard with a 1% chance of happening in any given year (1% AEP) based on the event- and response-based approaches. We find that the 1% AEP water depth can be produced by different events in different parts of the city and that the AEPs of these events are often*

*much larger than 1%. This means that the relationship between the probability of the event and the probability of flooding does not follow a one-to-one relationship."*

- I appreciate that the authors have pushed their code to GitHub. However, I visited <https://github.com/CoRE-Lab-UCF/MACH-Compound-Flooding/tree/main/Scripts> and was not able to figure out how to reproduce the results in the paper. It would be helpful for them to include a README file with instructions, a `main` script that runs the analysis, clear instructions for setting up the data, etc. I also was not able to figure out how the SFINCS code was run or where the input files for it are. It might be helpful to have a colleague check for reproducibility, as I would likely not be able to reproduce the results of this work.

*"The reviewer is correct in noting that the final analysis codes have not yet been uploaded to GitHub. We are currently modifying the original scripts because the analyses were conducted using SFINCS outputs in a different file format than those provided in the Zenodo repository. Due to file size constraints, we converted the original SFINCS simulation outputs to a more compact NetCDF format before uploading them to Zenodo. Given that we do not anticipate users attempting to rerun all 5,000 SFINCS simulations and subsequent downscaling to 1-meter subgrid resolution, we are updating the analysis scripts to work directly with the NetCDF files available on Zenodo. Once these modifications are complete, we will ask a colleague to test the full reproducibility of the workflow, as recommended by the reviewer, and add a readme file to guide users.*

---

## Author Response (AR2)

REVIEWER 1

The manuscript is much improved and the authors have addressed my comments in sufficient detail. Prior to final submission, I would encourage the authors to revisit the introduction again to streamline some of the text (see comments below). In addition, in the latter half of the paper, I notice that the authors replaced "AEP" with annual chance of occurring in any given year throughout. While I recognize that this was in response to the other reviewer's comments, and defer to the editor to make a final decision about the terminology, this change feels wordy and somewhat unnecessary once the meaning of AEP has been clarified. The change is especially tortuous in the first paragraph of the Results section.

*We thank the reviewer for the insightful comments, which have helped improve the manuscript. We agree that using AEP terminology is more concise; however, we also aimed to incorporate another reviewer's suggestion to avoid probabilities in certain sections to facilitate understanding for readers less familiar with the topic. We acknowledge that these changes may have reduced clarity in some places, so we have revised the manuscript (particularly the first paragraph of the Results section) to consistently use AEP terminology again.*

Minor technical comments

Line 47 should read "In the U.S., the event-based approach has been widely used and is currently recommended" FEMA's Future of Flood Risk Data (FFRD) Initiative would move away from the event-based approach and towards a response-based approach and is already under way, see, e.g., https://www.hec.usace.army.mil/confluence/hecnews/spring-2023/fema-s-future-of-flood-risk-data-initiative

*We thank the reviewer for pointing this out. We have revised the sentence to: "In the U.S., the event-based approach has been widely used by the Federal Emergency Management Agency (FEMA) to produce the 1% AEP flood elevations for both coastal and inland flood mapping, which serve as the basis for regulatory floodplain management and planning (FEMA, 2022)."*

*In addition, we have added the following sentence to line 95:*

*"For inland regions, FEMA is working to develop a methodology to transition to response-based (probabilistic) estimates."*

Line 50-55 This is one approach that is used for inland flooding, but I believe that river floodplains are also (deterministically) mapped using the return periods of water levels at gages.

*We have added the following line to clarify this point "In some cases, inland flooding is instead mapped using the 1% AEP river discharge estimated from stream gauges."*

Line 59 JMP should be JPM

*Corrected*

Lines 75-78 This sentence repeats some of the information provided in the previous paragraph

*This sentence refers specifically to rainfall and river discharge, whereas the previous paragraph focused on coastal water levels. We agree that similar approaches have been used for both inland and coastal drivers, and that this may sound somewhat repetitive, but we chose to retain this sentence to emphasize that the method has been applied across all flood drivers.*

Lines 100 "However, FEMA has not planned..." I don't think this statement is true and I would remove it in light of on-going work that is part of FFRD and related activities.

*Following the response to a previous comment, we have added: 'For inland regions, FEMA is working to develop a methodology to transition to response-based (probabilistic) estimates.' However, we have not found any FEMA activities specifically addressing 'compound flooding' or a response-based approach for coastal compound flooding. We do acknowledge, though, that FEMA is actively working to transition to a response-based approach for inland regions.*

Line 125 the k in km2 should be lowercase

*Corrected*

Line 530-531 It would be worth citing some hydrodynamic studies where the drivers were generated, perhaps from Gori and Lin 2022, Grimley et al., 2025 (preprint), or Bartlett et al., 2025 (preprint), to support this statement

https://agupubs.onlinelibrary.wiley.com/doi/full/10.1029/2022EF003097

https://essopenarchive.org/doi/full/10.22541/essoar.176030870.06323732

https://arxiv.org/pdf/2511.03871

*We thank the reviewer for the suggested references, which have been added to the relevant sentence.*

Line 535 It would make sense to cite USACE or FEMA's work on FFRD here as well

*Added*

REVIEWER 2

This is a strong paper and I am happy to recommend for publication, pending some minor improvements that in my view do not require review. Most of my original comments were suggestions for communication rather than deep critiques of the paper, and the authors have responded thoughtfully (often differently from how I would have done, but it is their paper not mine!)

*We thank the reviewer for the constructive comments and suggestions that have helped improve the manuscript, as well as for the kind words regarding our work.*

MAJOR COMMENTS
1. I was concerned that the paper "buried the lede". The authors have addressed this.

*Thanks*
2. I was concerned about level of polish of figures. Generally the figures are improved, but a few (eg, 8) are still blurry. I wonder if figure 2 could have subplots rearranged to fit better on the page? I would make (b)-(d) smaller and (a) larger.

*We have revised Figure 2 as suggested by the reviewer, changing the layout from vertical to horizontal. Our initial choice of a vertical layout was based on the journal's two-column format; however, given that the other reviewer also recommended changes to this figure in the first review, we have now enlarged panel (a) and repositioned panels (b–d) to the right side in a smaller format.*

*Regarding the map figures, we agree with the reviewer that the basemap quality was not optimal. Unfortunately, higher-resolution basemaps are not freely available. Public basemaps from ESRI (and other providers such as OpenStreetMap) do not offer high zoom levels without a paid license. We have therefore switched to a different publicly available ESRI basemap that is less colorful and maintains better clarity at the given zoom level. The basemaps for Figures 4, 7, and 8 have been updated accordingly.*

3. I disliked the use of probabilities / return periods to communicate compound events. The authors have addressed the comment appropriately -- I stand by my recommendation but where they disagree, they do so thoughtfully

*We thank the reviewer for this comment. However, based on the other reviewer's feedback regarding the use of 'AEP,' we have revised several sections again. We decided to revert to AEP terminology in some sentences while retaining the phrasing '1% chance of occurring in any given year' in parentheses to aid readers who may be less familiar with probabilistic terminology.*

4. I suggested that they better discuss similarities and differences between their hazard-generation framework and others; they have done an excellent job

*We thank the reviewer for this comment. We are pleased to have addressed this constructive suggestion, which we believe has strengthened the manuscript.*

Minor points are all addressed. I encourage the authors to tighten the figures a little bit and then I am excited for the community to read this paper.